Corrected: Author correction

# The genetic prehistory of the Baltic Sea region

Alissa Mittnik [1,2], Chuan-Chao Wang[1,3], Saskia Pfrengle[2], Mantas Daubaras[4], Gunita Zariņa[5], Fredrik Hallgren [6], Raili Allmäe[7], Valery Khartanovich[8], Vyacheslav Moiseyev[8], Mari Tõrv[9], Anja Furtwängler[2], Aida Andrades Valtueña[1], Michal Feldman[1], Christos Economou[10], Markku Oinonen [11], Andrejs Vasks[5], Elena Balanovska[12], David Reich[13,14,15], Rimantas Jankauskas [16], Wolfgang Haak[1,17], Stephan Schiffels [1] & Johannes Krause [1,2]

While the series of events that shaped the transition between foraging societies and food producers are well described for Central and Southern Europe, genetic evidence from Northern Europe surrounding the Baltic Sea is still sparse. Here, we report genome-wide DNA data from 38 ancient North Europeans ranging from ~9500 to 2200 years before present. Our analysis provides genetic evidence that hunter-gatherers settled Scandinavia via two routes. We reveal that the first Scandinavian farmers derive their ancestry from Anatolia 1000 years earlier than previously demonstrated. The range of Mesolithic Western hunter-gatherers extended to the east of the Baltic Sea, where these populations persisted without gene-flow from Central European farmers during the Early and Middle Neolithic. The arrival of steppe pastoralists in the Late Neolithic introduced a major shift in economy and mediated the spread of a new ancestry associated with the Corded Ware Complex in Northern Europe.

[1] Department of Archaeogenetics, Max Planck Institute for the Science of Human History, 07745 Jena, Germany. [2] Institute for Archaeological Sciences, Archaeo- and Palaeogenetics, University of Tübingen, 72070 Tübingen, Germany. [3] Department of Anthropology and Ethnology Xiamen University 361005 Xiamen, China. [4] Department of Archaeology, Lithuanian Institute of History, 01108 Vilnius, Lithuania. [5] Institute of Latvian History University of Latvia Riga, LV-1050, Latvia. [6] The Cultural Heritage Foundation, 72212 Västerås, Sweden. [7] Archaeological Research Collection, Tallinn University, 10130 Tallinn, Estonia. [8] Peter the Great Museum of Anthropology and Ethnography (Kunstkamera) RAS St. Petersburg, Russia 199034. [9] Institute of History and Archaeology, University of Tartu, 50090 Tartu, Estonia. [10] Archaeological Research Laboratory, Stockholm University, 11418 Stockholm, Sweden. [11] Finnish Museum of Natural History - LUOMUS, University of Helsinki, 00014 Helsinki, Finland. [12] Research Centre for Medical Genetics, Moscow, 115478, Russia. [13] Department of Genetics, Harvard Medical School, Boston, MA 02115, USA. [14] Broad Institute of Harvard and MIT, Cambridge, MA 02142, USA. [15] Howard Hughes Medical Institute, Harvard Medical School, Boston, MA 02115, USA. [16] Department of Anatomy, Histology and Anthropology, Vilnius University, 03101 Vilnius, Lithuania. [17] School of Biological Sciences The University of Adelaide Adelaide, SA 5005, Australia. Correspondence and requests for materials should be addressed to A.M. (email: mittnik@shh.mpg.de) or to J.K. (email: krause@shh.mpg.de)

Recent studies of ancient human genomes have revealed a complex population history of modern Europeans involving at least three major prehistoric migrations[1–6], influenced by climatic conditions, the availability of resources, the spread of technological and cultural innovations, and possibly diseases[7,8]. However, the archaeological record of the very north of the European subcontinent surrounding today's Baltic Sea shows a history distinct to that of Central and Southern Europe which has not yet been comprehensively studied on a genomic level.

Settlement of the Eastern Baltic and Scandinavia by mobile foragers started after the retreat of the glacial ice sheets around 11,000 years before present[9]. To the west and south, hunter-gatherers sharing a common genetic signature (Western Hunter-Gatherers or WHG; Supplementary Note 1 provides a glossary of abbreviations and archaeological terms) already occupied wide ranges of Europe for several millennia[1,2,5,10,11]. From further to the east, in the territory of today's Russia, remains of Mesolithic foragers have been studied (Eastern Hunter-Gatherers or EHG)[2,4]. They derived part of their ancestry, referred to as Ancient North Eurasian (ANE) ancestry, from a population related to the Upper Palaeolithic Mal'ta boy found in Siberia (MA1)[6,12]. Late Mesolithic foragers excavated in central Sweden, which have been called Scandinavian Hunter-Gatherers (SHG)[1,2], were modelled as admixed between WHG and EHG[6]. Archaeological evidence for the settlement of Scandinavia suggests both a route through southern Scandinavia and a route along the northern coast of Fennoscandia[13]. Foraging groups that inhabited the eastern coast and larger islands of the Baltic Sea as well as the Eastern Baltic inland during the 8th and 7th millennium calibrated radiocarbon years before Common Era (calBCE) developed a dual habitation system, establishing more permanent settlements than their surrounding contemporaries while remaining partially mobile[14,15].

The following Early Neolithic period, starting around 6000 calBCE, saw the transition from foraging to a sedentary agricultural lifestyle with the expansion of farmers out of Anatolia into Central and Southern Europe[1,4,6,16,17]. This development reached southern Scandinavia at around 4000 calBCE with farmers of the so-called Early Neolithic Funnel Beaker Culture (EN TRB; from German *Trichterbecher*) who gradually introduced cultivation of cereals and cattle rearing. At the transition to the northern Middle Neolithic, around 3300 calBCE, an intensification of agriculture occurred in Denmark and in western central Sweden accompanied by the erection of megaliths. Settlements in eastern central Sweden increasingly concentrated along the coast, where the economy shifted towards the marine resources. Early pottery of these coastal hunter-gatherers, known as the Pitted Ware Culture (PWC), resembles the Funnel beakers in shape. Analysis of ancient genomes from PWC and megalithic Middle Neolithic TRB (MN TRB) context in central Sweden has shown that the PWC individuals retain the genetic signature of Mesolithic hunter-gatherers while the TRB farmers' ancestry can mainly be traced back to Central European farmers, albeit with substantial admixture from European hunter-gatherers[18–20]. As these TRB individuals date to a period one millennium after the initial Neolithization in southern Scandinavia, the question remains whether the first introduction of farming around 4000 BCE was driven by newcomers or by local groups involving later gene-flow from Central European farmers.

The production and use of pottery, in Central and Southern Europe often seen as part of the 'Neolithic package', was already common among foragers in Scandinavia during the preceding Mesolithic Ertebølle phase. Similarly, in the Eastern Baltic, where foraging continued to be the main form of subsistence until at least 4000 calBCE[15], ceramics technology was adopted before agriculture, as seen in the Narva Culture and Combed Ceramic Culture (CCC). Recent genome-wide data of Baltic pottery-producing hunter-gatherers revealed genetic continuity with the preceding Mesolithic inhabitants of the same region as well as influence from the more northern EHG[21,22], but did not reveal conclusively whether there was a temporal, geographical or cultural correlation with the affinity to either WHG or EHG.

The transition from the Late (Final) Neolithic to the Early Bronze Age (LNBA) is seen as a major transformative period in European prehistory, accompanied by changes in burial customs, technology and mode of subsistence as well as the creation of new cross-continental networks of contact seen in the emergence of the pan-European Corded Ware Complex (CWC, ca. 2900–2300 calBCE) in Central[2] and north-eastern Europe[21]. Studies of ancient genomes have shown that those associated with the CWC were closely related to the pastoralists of the Yamnaya Culture from the Pontic-Caspian steppe, introducing a genetic component that was not present in Europe previously[2,3]. This genetic component is hypothesized to have spread in the subsequent millennia throughout Europe and can be seen in today's European populations in a decreasing north-east to south-west gradient.

Intriguingly, modern Eastern Baltic populations carry the highest proportion of WHG ancestry of all Europeans[1], supporting the theory that the hunter-gatherer population of this region left a lasting genetic impact on subsequent populations[23].

Here, we investigate the modes of cultural and economic transitions experienced by the prehistoric populations surrounding the Baltic Sea. Were the changes seen in the Eastern Baltic Neolithic, which did not involve the introduction of agriculture, driven by contact with neighbouring groups and if so can we identify these? Was the earliest practice of farming in southern Scandinavia a development by a local population or did it involve migration from the South? How did the unique genetic signature of modern Eastern Baltic populations come to be?

We present novel genome-wide data from 38 ancient individuals from the Eastern Baltic, Russia and Sweden spanning 7000 years of prehistory, covering the transition from a mobile hunter-gatherer to a sedentary agricultural lifestyle, as well as the adoption of bronze metallurgy. We show that the settlement of Scandinavia by hunter-gatherers likely took place via at least two routes, and that the first introduction of farming was brought about by the movement of the Central European farmers into the region at around 4000 calBCE. In the Eastern Baltics, foraging remained the dominant economy among interconnected north-eastern hunter-gatherer groups that did not experience admixture from European farmers until around 3000 calBCE, when a shift towards agro-pastoralism came about through migrations from the Pontic-Caspian steppe.

## Results

**Samples and archaeological background.** The skeletal remains studied here were recovered from 25 archaeological sites in the territory of modern Lithuania, Latvia, Estonia, Archangelsk Oblast and Karelia (north-western Russia) and Sweden dating from around 7500 to 200 calBCE (Fig. 1, Supplementary Note 2, Supplementary Data 1). In total, we analyzed DNA from 106 human remains. A total of 41 samples with good DNA preservation were selected for deeper shotgun sequencing or SNP capture (Supplementary Data 1). In the latter case, we enriched samples for a panel of around 1.24 million single nucleotide polymorphisms (SNPs) via in-solution capture[4,24]. After quality control, genome-wide data from 38 individuals, with an average coverage of 0.02–8.8-fold on targeted SNPs, were included in further analysis.

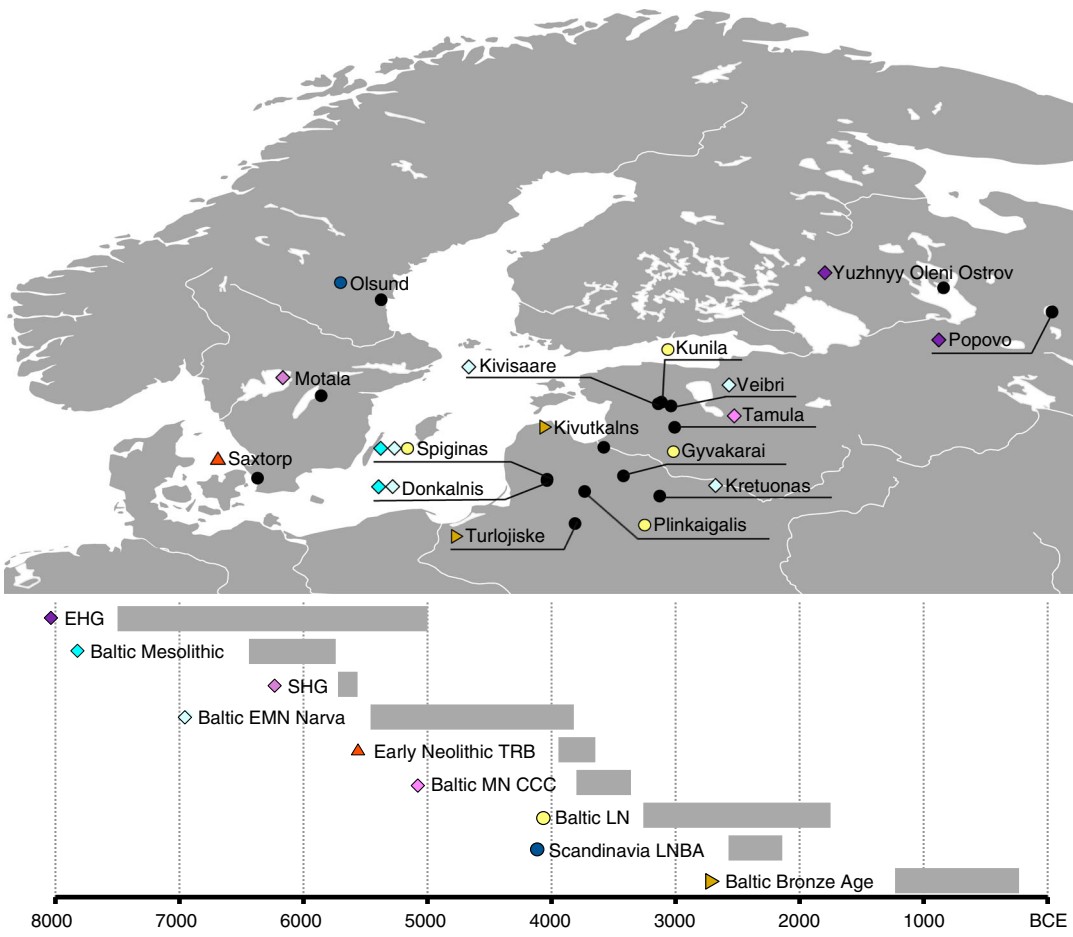

**Fig. 1** Sampling locations and dating of 38 ancient Northern European samples introduced in this study. Chronology based on calibrated radiocarbon dates or relative dating, see Supplementary Note 2. Map generated with QGIS 2.18.2 (http://www.qgis.org/) using the Natural Earth data set (http://www.naturalearthdata.com) for the basemap

The 38 final samples fall into nine broad groups (Table 1): first, two Mesolithic hunter-gatherers from north-western Russia (ca. 7500–5000 calBCE); secondly, an individual (5720–5560 calBCE) from the lake burial site of Motala, Sweden, adding to the previously published six SHG individuals from this site[4]; two Mesolithic hunter-gatherers from Lithuania (ca. 6440–5740 calBCE) associated with the Kunda Culture (referred to as Baltic Mesolithic), whose archaeological assemblages found in Lithuania, Latvia, Estonia and adjacent parts of Russia.

Twelve individuals were associated with pottery-producing forager cultures. Ten of them from Lithuania and Estonia (ca. 5460–3820 calBCE) were assigned to the Narva Culture that occupied the Eastern Baltic region from the Late Mesolithic to the Middle Neolithic (Baltic EMN Narva) and two individuals from Estonia were associated with the CCC that was spread across the northern part of the Eastern Baltic by the Middle Neolithic (Baltic MN CCC; dated to ca. 3800–3360 calBCE).

Five individuals from Lithuania and Estonia were dated to the Late Neolithic (Baltic LN; ca. 3260–1750 calBCE). For the individual Gyvakarai1, we present genome-wide data at 7.6-fold average coverage from shotgun sequencing.

Fourteen samples from Latvia and Lithuania were attributed to the Baltic Bronze Age (Baltic BA) and date to ca. 1230–230 calBCE.

We supplemented our dataset of ancient Eastern Baltic samples with recently published data from 13 individuals spanning the Mesolithic to Late Neolithic in Latvia[21] and Estonia[22], and merged data from identical individuals where they overlapped

with the latter. We present the first data from the southernmost region of the Eastern Baltic (the territory of modern-day Lithuania), the Early Neolithic of Estonia and the Eastern Baltic Bronze Age.

From southern Sweden, we analyzed one farmer (3950–3650 calBCE) from the EN TRB, the earliest agricultural population in Scandinavia for which there exists no genetic data to date. One sample from northern Sweden (Scandinavia LNBA Olsund, ca. 2570–2140 calBCE) dates to the Late Neolithic but was found without associated archaeological assemblages. The data were analyzed in context with published data from the Scandinavian Middle Neolithic to Bronze Age[3,19] as well as other ancient and modern genome-wide datasets described below.

**Affinities of northern Mesolithic hunter-gatherers**. To gain an overview of the broad genetic affinities of our samples, we projected all 38 ancient genome-wide datasets, as well as previously published ancient samples[4,6,19,21,22], using a principal component analysis (PCA), constructed from 1007 modern individuals from a diverse set of West Eurasian contemporary populations, and used the same individuals to investigate model-based clustering using ADMIXTURE. We see that the Mesolithic foragers of Northern Europe fall into three distinct clusters, associated with EHG, SHG and WHG, respectively, as evidenced by their position on the PCA (Fig. 2a), similar composition of ancestral genetic clusters in ADMIXTURE analysis (Fig. 2b, Supplementary Fig. 4) and in sharing most genetic drift since divergence from Africa as

**Table 1 Information on ancient samples for which we report the nuclear data in this study**

| Sample name | 95.4% CI calibrated radiocarbon age (calBCE)/contextual dating (BCE) | Population label | Site location | Genetic sex | SNPs overlapping 1240k set | Average coverage on 1240k SNPs | mtDNA haplogroup | Y-haplogroup* |
|---|---|---|---|---|---|---|---|---|
| UzOO77 | 5500–5000 BCE | EHG | Yuzhnyy Oleni Ostrov, Karelia, Russia | F | 530434 | 0.733 | R1b |  |
| Popovo2 | 7500–5000 BCE | EHG | Popovo, Archangelsk, Russia | M | 68042 | 0.064 | U4d | n/a |
| MotalaAA | 5722–5564 calBCE | SHG | Kanaljorden, Motala, Sweden | F | 56508 | 0.055 | U5a2d |  |
| Donkalnis4 | 6000–5740 calBCE | Baltic Mesolithic | Donkalnis, Lithuania | M | 22005 | 0.021 | U5b2c1 | I |
| Spiginas4 | 6440–6230 calBCE | Baltic Mesolithic | Spiginas, Lithuania | F | 663885 | 1.122 | U4a2 |  |
| Donkalnis1 | 5500/5300–3100/2900 BCE | Baltic EMN Narva | Donkalnis, Lithuania | F | 47228 | 0.045 | U5b1 |  |
| Donkalnis7 | 5460–4940 calBCE | Baltic EMN Narva | Donkalnis, Lithuania | M | 458738 | 0.758 | U5a2d1 | R1 |
| Veibri4 | 4900–4720 calBCE | Baltic EMN Narva | Veibri, Estonia | F | 542733 | 1.047 | U5b1 |  |
| Kivisaare3 | 4730–4540 calBCE | Baltic EMN Narva | Kivisaare, Estonia | M | 176533 | 0.186 | U4a1 | n/a |
| Spiginas1 | 4440–4240 calBCE | Baltic EMN Narva | Spiginas, Lithuania | M | 962584 | 6.106 | H11a | I2a1a2a1a |
| Donkalnis6 | 4720–4530 calBCE | Baltic EMN Narva | Donkalnis, Lithuania | F | 933997 | 6.030 | U5a2e |  |
| Kretuonas1 | 4460–3820 calBCE | Baltic EMN Narva | Kretuonas 1B, Lithuania | F | 297696 | 0.367 | U5b1 |  |
| Kretuonas5 | 4450–4340 calBCE | Baltic EMN Narva | Kretuonas 1B, Lithuania | M | 192523 | 0.204 | U5b2b | I |
| Kretuonas4 | 5500/5300–3100/2900 BCE | Baltic EMN Narva | Kretuonas 1B, Lithuania | F | 993319 | 8.792 | U5b1b1a |  |
| Kretuonas2 | 5500/5300–3100/2900 BCE | Baltic EMN Narva | Kretuonas 1B, Lithuania | M | 634269 | 1.282 | U5b2b | I2a1b |
| Tamula1 | 3630–3360 calBCE | Baltic MN CCC | Tamula, Estonia | F | 160270 | 0.168 | U5a1d2b |  |
| Tamula3 | 3800–3640 calBCE | Baltic MN CCC | Tamula, Estonia | M | 153219 | 0.160 | U4d2 | R1 |
| Saxtorp5164 | 3945–3647 calBCE | EN TRB | Kvärlöv, Saxtorp, Skåne, Sweden | F | 370367 | 0.587 | T2b |  |
| Kunila2** | 2580–2340 calBCE | Baltic LN | Kunila, Estonia | M | 382562 | 0.455 | J1c3 | R1a1a1 |
| Gyvakarai1 | 2620–2470 calBCE | Baltic LN | Gyvakarai, Lithuania | M | 1096987 | 7.559 | K1b2a | R1a1a1b |
| Spiginas2 | 2130–1750 calBCE | Baltic LN | Spiginas, Lithuania | M | 870598 | 3.164 | I4a | R1a1a1b |
| Plinkaigalis242 | 3260–2630 calBCE | Baltic LN | Plinkaigalis, Lithuania | F | 861862 | 2.574 | W6a |  |
| Plinkaigalis241 | 2860–2410 calBCE | Baltic LN | Plinkaigalis, Lithuania | F | 190225 | 0.213 | I2 |  |
| Olsund | 2573–2140 calBCE | Scandinavia LNBA | Ölsund, Hälsingland, Sweden | M | 682911 | 2.225 | U4c2a | R1a1a1b |
| Turlojiske1 | 930–810 calBCE | Baltic BA | Turlojiškė, Lithuania | M | 127416 | 0.131 | T2b | R1a1a1b |
| Turlojiske3 | 1010–800 calBCE | Baltic BA | Turlojiškė, Lithuania | M | 471779 | 0.671 | H4a1a1a3 | R1a1a1b |
| Turlojiske5 | 2100/2000–600 BCE | Baltic BA | Turlojiškė, Lithuania | M | 59416 | 0.058 | H5 | CT |
| Turlojiske1932 | 1230–920 calBCE | Baltic BA | Turlojiškė, Lithuania | F | 82860 | 0.081 | U5a2a1 |  |
| Kivutkalns153 | 800–545 calBCE | Baltic BA | Kivutkalns, Latvia | M | 246417 | 0.334 | U5a1a1 | R1b1a2 |
| Kivutkalns164 | 730–390 calBCE | Baltic BA | Kivutkalns, Latvia | M | 95172 | 0.093 | U5a2a1 | R1a1 |
| Kivutkalns19 | 730–400 calBCE | Baltic BA | Kivutkalns, Latvia | M | 896471 | 5.760 | H10a | R1a1a1b |
| Kivutkalns25 | 800–545 calBCE | Baltic BA | Kivutkalns, Latvia | M | 682042 | 1.569 | H28a | R1a1a1b |
| Kivutkalns42 | 810–560 calBCE | Baltic BA | Kivutkalns, Latvia | F | 585203 | 1.102 | H1b1 |  |
| Kivutkalns194 | 800–545 calBCE | Baltic BA | Kivutkalns, Latvia | M | 130958 | 0.152 | T1a1b | R1a1a |
| Kivutkalns207 | 730–390 calBCE | Baltic BA | Kivutkalns, Latvia | F | 915334 | 7.212 | H1b2 |  |
| Kivutkalns209 | 405–230 calBCE | Baltic BA | Kivutkalns, Latvia | M | 807138 | 2.240 | J1b1a1 | R1a1a |
| Kivutkalns215 | 790–535 calBCE | Baltic BA | Kivutkalns, Latvia | F | 850417 | 2.735 | H1c |  |
| Kivutkalns222 | 805–515 calBCE | Baltic BA | Kivutkalns, Latvia | M | 641886 | 1.278 | U5a1c1 | R1a1 |

Radiocarbon dates for Spiginas1 and Donkalnis7 were first reported in ref. [28], radiocarbon dates for Spiginas2, Donkalnis6, Kretuonas5, Gyvakarai, and Turlojiškė1 were first reported in ref. [30]
*Y-haplogroups are based on the most downstream defining mutation covered and might not reflect the true haplogroup, see Supplementary Note 3
**Data merged with published data from this individual [22]

shown by outgroup $f_3$-statistics (Supplementary Fig. 1). Based on these results, our Mesolithic Russian foragers fall within the EHG cluster formed by previously published samples[4] and are grouped as EHG in subsequent analyses.

ADMIXTURE shows that EHG carry a genetic component (green component in Fig. 2b) that is maximized in hunter-gatherers from the Caucasus (CHG) and shared with Neolithic farmers from Iran and Steppe populations from the Bronze Age, suggesting some common ancestry for these populations, consistent with previous results[21].

Despite their geographical vicinity to EHG, the two Eastern Baltic individuals associated with the Mesolithic Kunda Culture show a very close affinity to WHG in all our analyses (Fig. 2, Supplementary Figs. 1 and 2), with a significant contribution from ANE, as revealed by negative admixture $f_3$ results involving a Palaeolithic hunter-gatherer from Switzerland, most closely related to WHG, and populations containing ANE ancestry (Supplementary Table 1). This is additionally confirmed by $D$-statistics of the form $D$(Baltic Mesolithic, WHG; $X$, Mbuti) for populations $X$ with ANE ancestry, which are significant and among the highest in EHG ($Z = 6.2$; Supplementary Table 2).

Using the $qpWave/qpAdm$ framework, we modelled the Baltic Mesolithic hunter-gatherers as a two-way mixture between EHG and WHG (Fig. 3), which reveals a difference in mixture proportions between the more northern individuals from the Latvian site[21] (65–76% WHG with 24–35% EHG; Supplementary Table 3) and the samples from the Lithuanian sites to the south (88–100% WHG with 0–12% EHG; Supplementary Table 3).

SHG appear intermediate between WHG/Baltic Mesolithic and EHG in PCA space, have increased shared genetic drift with both shown in outgroup $f_3$-statistics (Supplementary Fig. 3) and the statistic $D$(SHG,WHG; EHG, Mbuti) is strongly significant for excess allele sharing of SHG and EHG ($Z = 7.3$; Supplementary Table 4). Using $qpAdm$, we confirm the previously published result of SHG being formed by admixture of WHG and EHG[6] ($57 \pm 2\%$ WHG with $43 \pm 2\%$ EHG; $p = 0.19$; Supplementary Table 3). Both EHG and SHG share a non-negligible component in ADMIXTURE analysis that is maximized in some modern Native American populations which points towards ANE ancestry, as represented by the MA1 and AG3 samples from Palaeolithic Siberia[12] (maroon component in Fig. 2a, Supplementary Fig. 4). Indeed, $D$-statistics show that EHG and SHG

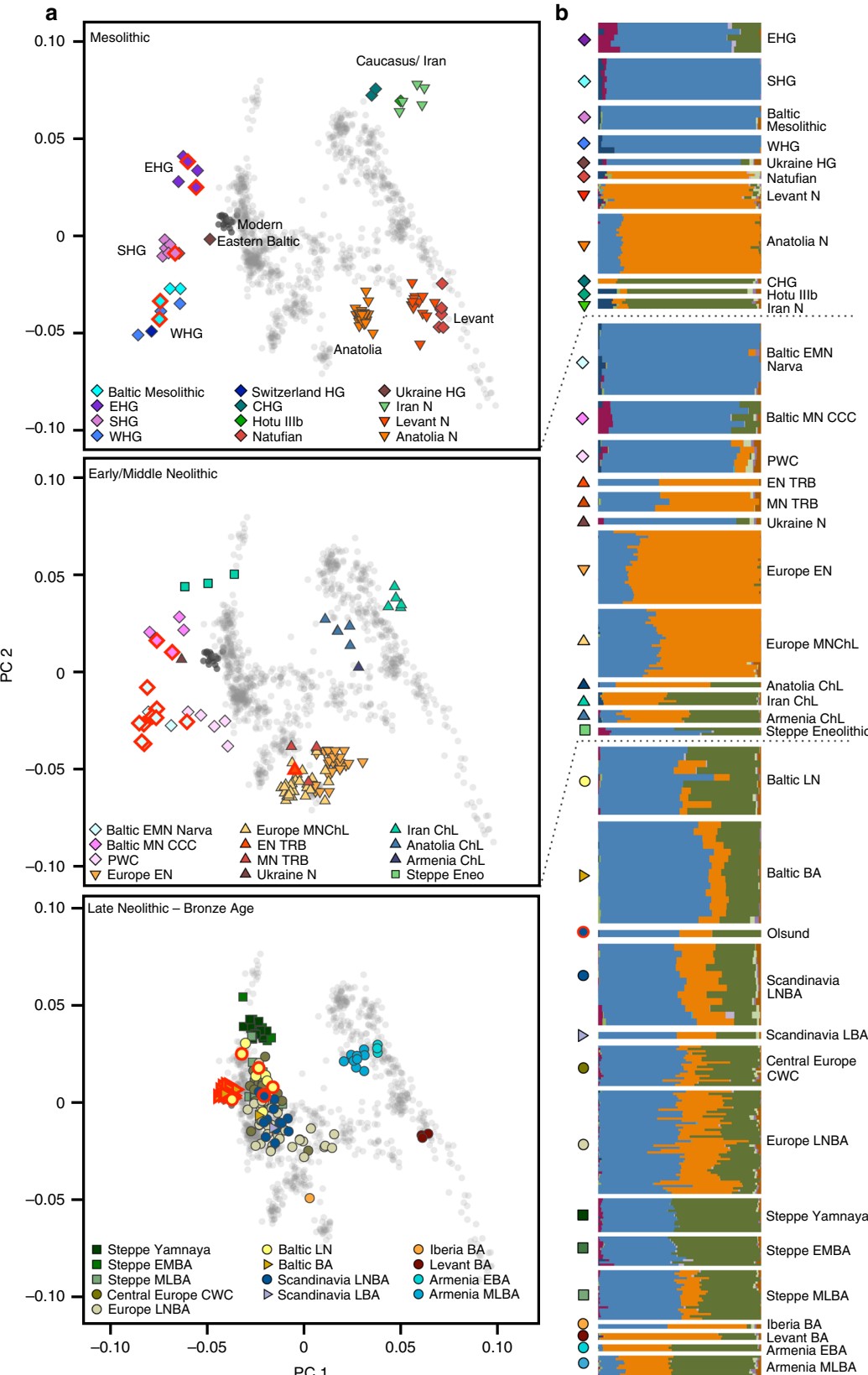

**Fig. 2** PCA and ADMIXTURE analysis reflecting three time periods in Northern European prehistory. **a** Principal components analysis of 1012 present-day West Eurasians (grey points, modern Baltic populations in dark grey) with 294 projected published ancient and 38 ancient North European samples introduced in this study (marked with a red outline). Population labels of modern West Eurasians are given in Supplementary Fig. 7 and a zoomed-in version of the European Late Neolithic and Bronze Age samples is provided in Supplementary Fig. 8. **b** Ancestral components in ancient individuals estimated by ADMIXTURE ($k = 11$)

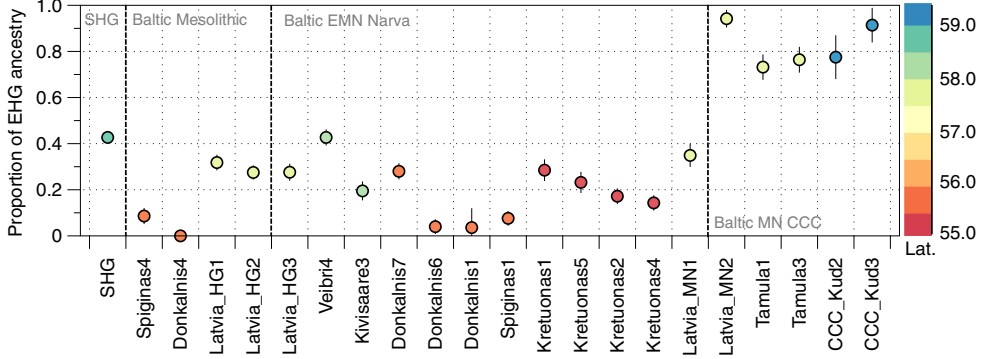

**Fig. 3** EHG ancestry in SHG and Eastern Baltic hunter-gatherers. Ancestry proportions were estimated with *qpAdm*, and standard errors are shown as vertical lines. Colours indicate latitude of the site at which the individuals were excavated

share significantly more alleles with MA1 and AG3 than WHG (Supplementary Tables 2 and 4). Additionally, mtDNA haplogroups found among EHG point towards an eastern influence (Table 1): R1b in UzOO77 (this individual had previously been assigned to haplogroup H[58]) was also found in the Palaeolithic Siberian AG3[5] and a haplogroup within the C1 clade, which appears today in highest frequencies in north-east Asia and the Americas, was described in several samples from Yuzhnyy Oleni Ostrov[25,26]. It was shown that some SHG carry the derived variant of the *EDAR* allele, which affects hair thickness and tooth morphology among other things, and which is found today in high frequency in East Asians and Native Americans[4].

**Dynamic forager networks in the Eastern Baltic Neolithic.** Similarly to the Baltic Mesolithic, the later Eastern Baltic Neolithic hunter-gatherers of the Narva culture exhibit varying proportions of EHG (0–46%) and WHG (54–100%) ancestry (Fig. 3 and Supplementary Table 3). In principle, it is possible that such a pattern is not the result of admixture but a signal of a long-standing geographic cline. However, in this case, it appears to be more compatible with recent admixture between differentially WHG- and EHG-related groups, as we see varying ancestry proportions even within contemporary individuals from the same site or closely located sites (Fig. 3). *D*-statistics of the form *D* (Baltic EMN Narva, WHG; *X*, Mbuti) do not show evidence of admixture with the contemporaneous European farming populations that were related to the Anatolian Neolithic (Supplementary Tables 2 and 5). The later individuals attributed to the Baltic MN CCC exhibit a significantly higher affinity to EHG with the ancestry proportion estimated at 68–99% EHG and 1–32% WHG (Fig. 3, Supplementary Table 3).

Similar to other European Mesolithic hunter-gatherers, our Baltic foragers carry a high frequency of the derived *HERC2* allele which codes for light iris colour, and like SHG and EHG they already possess an increased frequency of the derived alleles for *SLC45A2* and *SLC24A5*, coding for lighter skin colour (Supplementary Table 6). The male individuals carry Y-chromosomal haplogroups of the I and R1 clades (Table 1, Supplementary Note 3). Y-haplogroup I has been most commonly found among WHG and SHG[1,5] while R1 is found in EHG[2] and other published Eastern Baltic[21,22] and Romanian hunter-gatherers[27].

One Narva individual, Spiginas1[28], dated to ca. 4440–4240 calBCE, belongs to a mitochondrial haplogroup of the H branch, normally associated with the Neolithic expansion into Europe, but shows no evidence of Neolithic farmer ancestry on the nuclear level suggesting that this haplogroup might have been present already in foraging groups (Table 1, Supplementary Table 5). In addition to haplogroup H, the maternal lineages seen in Eastern Baltic samples (*n* = 35; Supplementary Fig. 5)

encompass all of the major haplogroups identified in complete mtDNA genomes from Holocene Scandinavian and Western European hunter-gatherers (*n* = 21:U2, U5a, U5b)[29], as well as haplogroup U4 which has been found in high frequency in Mesolithic foragers from Russia[25]. We also find mtDNA branch K1, a subclade of U8, in one Baltic Mesolithic forager, adding to the mounting evidence that this lineage was present at low frequency among European hunter-gatherers before the arrival of agriculturalists[16,27].

**Early farming in Sweden coincides with a shift in ancestry.** In contrast to the Eastern Baltic, we see clear evidence for the genetic impact of the Neolithic expansion already around 4000 calBCE in southern Sweden. The individual associated with the EN TRB culture clusters with Middle and Early Neolithic farmers from Europe on the PCA (Fig. 2a) and in the ADMIXTURE analysis exhibits the component maximized in Levantine and West Anatolian early farmers (orange component in Fig. 2b). *D*-statistics show increased allele sharing of EN TRB with hunter-gatherer populations in comparison to the preceding farmers of the *Linearbandkeramik* (LBK) culture and no difference to the contemporaneous Middle Neolithic and Chalcolithic (MNChL) farmers of Central Europe (Supplementary Table 4), paralleling the previously described resurgence of WHG ancestry during the European Middle Neolithic[2,27]. Different models of EN TRB as a linear combination of LBK with SHG, WHG or Baltic hunter-gatherers favour the latter two groups, while SHG is rejected as a source (Supplementary Table 3).

The previously published succeeding farmers of the Middle Neolithic (MN) TRB culture in West Sweden[19] appear as directly descended from the EN TRB, with no significant positive results for *D*(MN TRB, EN TRB; *X*, Mbuti) (Supplementary Table 4).

The PWC individuals, who were contemporaneous to the MN TRB but relied mainly on marine resources, appear intermediate between SHG and Middle Neolithic farming cultures on the PCA (Fig. 2a). Indeed, the statistic *D*(PWC, SHG; *X*, Mbuti) reaches weak significance when *X* is MN TRB (*Z* = 2.94) and a two-way admixture model for PWC involving SHG and TRB farmers is not rejected by *qpWave/qpAdm* (74 ± 6% SHG and 26 ± 6% EN TRB; Supplementary Table 3).

**New networks of contact during the LNBA.** The substantial population movement at the beginning of the 3rd millennium calBCE, during the European LNBA, affected the genetic makeup of Eastern and Central Europe and Scandinavia[2,3]. It also made its mark in the Eastern Baltic region, as seen in our five samples from Lithuania and Estonia dated to this period and the previously published individuals from the Eastern Baltic region[21,22].

All Baltic LN individuals (ca. 3200–1750 calBCE) fall in PCA space in the diffuse European LNBA cluster formed by individuals admixed between Early and Middle Bronze Age pastoralists from the Yamnaya culture of the eastern Pontic Steppe and Middle Neolithic European farmers (Fig. 2a). They all carry the genetic component that was introduced into Europe with this pastoralist migration in varying amounts, and the majority also carries the component associated with Anatolian farmers (green and orange, respectively, in Fig. 2b). This genetic impact is furthermore reflected in the uniparental markers where we see novel mitochondrial haplogroups (I, J, T2, W), not found in the preceding foragers, in half of our samples (Supplementary Fig. 5), and I2a Y-chromosomal haplogroups replaced by R1a types (Table 1, Supplementary Note 3).

Computing $D$-statistics for each individual of the form $D$(Baltic LN, Yamnaya; $X$, Mbuti), we find that the two individuals from the early phase of the LN (Plinkaigalis242 and Gyvakarai1, dating to ca. 3200–2600 calBCE) form a clade with Yamnaya (Supplementary Table 7), consistent with the absence of the farmer-associated component in ADMIXTURE (Fig. 2b). Younger individuals share more alleles with Anatolian and European farmers (Supplementary Table 7) as also observed in contemporaneous Central European CWC individuals[2]. The individual Spiginas2[30], dated to a very late period of the LN (2130–1750 calBCE), stands out in that it shares an excess of alleles with European forager groups when compared to the Yamnaya populations, with the top hits being Switzerland_HG, WHG, Baltic Mesolithic and Baltic EMN Narva (Supplementary Table 7).

This result is the earliest evidence for a continuing pattern: we observe that the increased affinity to Baltic hunter-gatherers remains prevalent in the more recent samples from the Baltic BA (dated between ca. 1000 and 230 calBCE) that cluster together on the PCA in the same space occupied by modern Lithuanians and Estonians, shifted from other Europeans to WHG and Baltic hunter-gatherers (Fig. 2a). The statistic $D$(Baltic BA, Baltic LN; Baltic EMN Narva, Mbuti) is strongly significant ($Z = 14.0$; Supplementary Table 2) demonstrating the increase in allele sharing with local hunter-gatherers in the Baltic populations after the Late Neolithic. Replacing Baltic EMN Narva with the contemporaneous northern population of Baltic MN CCC does not yield significant results, suggesting that admixture with this population did not play a large role in the ancestry of our studied Bronze Age individuals. Additionally, $D$-statistics provide significantly positive results for $D$(Baltic BA, Baltic LN; $X$, Mbuti) when $X$ was replaced by various agricultural populations of Europe and the Near East (Supplementary Table 2), which suggests that the formation of the Baltic BA gene pool was not completed by admixture between the Baltic LN population and foragers but involved additional gene-flow from outside the Baltic territory. Archaeological evidence supports that the site Kivutkalns, which is represented by 10 of our individuals, was a major bronze-working centre located on a trade route that opened to the Baltic Sea on the west and led inland following the Daugava river[31], where contact to surrounding populations might have been common.

The individual from Olsund in north-eastern Sweden was dated to the Late Neolithic, when agriculture had been introduced to the coastal areas of northern Sweden with the Battle Axe Culture, the regional variant of the CWC, while foraging persisted as an important form of subsistence. The remains were found without any associated artefacts, but in close proximity to a site where the assemblage showed a mix between local hunter-gatherer traditions and CWC influence[32].

On the PCA plot, this sample falls within the European LNBA cluster (Fig. 2a) and similarly to other individuals from this cluster displays the three components derived from WHG, CHG and Neolithic Levant (Fig. 2b). This provides genomic evidence for the influence of both the early Neolithic and LNBA expansions having reached as far as northern Sweden in the 3rd millennium calBCE, either through a northward expansion from southern Scandinavia or across the Baltic Sea by boat or over the frozen sea[33]. Assemblages similar to the early Battle Axe Culture of Sweden have been found in south-western Finland, across the Bothnian Sea[34,35] which could be considered a geographically closer source than southern Scandinavia.

**Gene-flow into the Eastern Baltic after the Bronze Age.** Modern Eastern Baltic populations cluster with Baltic BA on the PCA plot and exhibit among all modern populations the highest shared genetic drift with ancient Baltic populations (Supplementary Fig. 2), but show substantial differences to samples from the Bronze Age. The statistic $D$(Lithuanian, Baltic BA; $X$, Mbuti) reveals significantly positive results for many modern Near Eastern and Southern European populations (Supplementary Fig. 6a). Limited gene-flow from more south-western neighbouring regions after the Bronze Age is sufficient to explain this pattern, as nearly all modern populations besides Estonians, especially for Central and Western Europe, have a higher amount of farmer ancestry than Lithuanians.

In contrast, the statistic $D$(Estonian, BA Baltic; $X$, Mbuti) gives significant positive hits for East Asian and Siberian populations (Supplementary Fig. 6b).

None of our male Bronze Age individuals carry Y-haplogroup N, which is found in modern Europeans in highest frequencies in Finland and the Baltic states[36]. Instead, we observe a high frequency of R1a Y-haplogroups.

**Discussion**

Our analyses support a dynamic population history of the Baltic Sea region, where populations did not remain in 'genetic stasis' despite the late adoption of agrarian subsistence strategies when compared to the rest of Europe.

The Mesolithic SHG excavated at Motala, Sweden, owe their genetic signature to an admixture of WHG and EHG and similarly to the latter carry substantial ANE ancestry. In contrast, the two Eastern Baltic Mesolithic Kunda individuals, who predate the SHG, carry comparatively low proportions of ANE ancestry, indicating that this ancestry was never widespread to the south-east of the Baltic Sea and likely reached Scandinavia without traversing the Eastern Baltic. This provides indirect support to the archaeological evidence that Scandinavia was settled by two routes[13], suggesting a scenario in which the ANE-related ancestry was brought into Scandinavia with a movement of people via a north-eastern coastal route, where they admixed with a WHG-like population that derived from a migration over the land-bridge that connected Denmark and southern Sweden at the time. This scenario is also supported by the finding that three Mesolithic hunter-gatherers excavated at the coast of Norway carry a higher proportion of EHG ancestry compared to the individuals from inland Sweden[37].

In southern Scandinavia, the sequence of events resembles that seen in Central Europe, albeit several millennia later, in that the earliest agriculture in the region coincides with the appearance of people related to the Anatolian and European Neolithic. However, similar to Middle Neolithic Central and Southern European populations, early Scandinavian farmers are already strongly admixed with hunter-gatherer groups. Without the knowledge of the genetic substrate in Mesolithic southern Scandinavia, these results are consistent with different scenarios; e.g. a movement of

a Central European population into southern Scandinavia without admixture with local SHG-like populations, or local admixture of LBK-like farmers with a forager population that shows more similarity to WHG or Baltic hunter-gatherers[19]. A detailed joint analysis of genetic and archaeological data from hunter-gatherers from northern Germany, Denmark and the southern tip of Sweden is necessary to establish the role of local admixture during the emergence of the TRB culture. Our data support that the Neolithic PWC foragers are largely genetically continuous to SHG, which is congruent with their similarities in subsistence strategies, while continuity between EN TRB and PWC can also be seen in archaeological assemblages[38] and can be attributed to contact between farmers and foragers. Indeed, genetic evidence of admixture between these groups shows that they were not completely isolated from each other but did likely not uphold continuous contact nor intermarry frequently during their prolonged parallel existence in Scandinavia.

In the archaeological understanding, the transition from Mesolithic to Neolithic in the Eastern Baltic region does not coincide with a large-scale population turnover and a stark shift in economy as seen in Central and Southern Europe. Rather, it is signified by a change in networks of contacts and the use of pottery, among other material, cultural and economic changes[15]. Our results suggest continued admixture between groups in the south of the Eastern Baltic region, who are more closely related to WHG, and northern or eastern groups, more closely related to EHG. Neolithic social networks from the Eastern Baltic to the River Volga could also explain similarities of the hunter-gatherer pottery styles, although morphologically analogous ceramics might also have developed independently due to similar functionality[39]. The genetic evidence for a change in networks and possibly even a large-scale population movement is most pronounced in the Middle Neolithic in individuals attributed to the CCC. The distribution of this culture overlaps in the north with the Narva culture and extends further north to Finland and Karelia. Its spread in the Eastern Baltic is linked with a significant change in imported raw materials, artefacts, and the appearance of village-like settlements[15].

We see a further population movement into the regions surrounding the Baltic Sea with the CWC in the Late Neolithic that was accompanied by the first evidence of extensive animal husbandry in the Eastern Baltic[15]. The presence of ancestry from the Pontic-Caspian Steppe among Baltic CWC individuals without the genetic component from north-western Anatolian Neolithic farmers must be due to a direct migration of steppe pastoralists that did not pick up this ancestry in Central Europe. It suggests import of the new economy by an incoming steppe-like population independent of the agricultural societies that were already established to the south and west of the Baltic Sea. The presence of direct contacts to the steppe could lend support to a linguistic model that sees an early branching of Balto-Slavic from a Proto-Indo-European language, for which the west Eurasian steppe was proposed as a homeland[40–42]. However, as farmer ancestry is found in later Eastern Baltic individuals, it is likely that considerable individual mobility and a network of contact throughout the range of the CWC facilitated its spread eastward, possibly through exogamous marriage practices[43]. Conversely, the appearance of mitochondrial haplogroup U4 in the Central European Late Neolithic after millennia of absence[44] could indicate female gene-flow from the Eastern Baltic, where this haplogroup was present at high frequency.

Local foraging societies were, however, not completely replaced and contributed a substantial proportion to the ancestry of Eastern Baltic individuals of the latest LN and Bronze Age. This 'resurgence' of hunter-gatherer ancestry in the local population through admixture between foraging and farming groups recalls the same phenomenon observed in the European Middle Neolithic[2,45] and is responsible for the unique genetic signature of modern-day Eastern Baltic populations.

We suggest that the Siberian and East Asian related ancestry in Estonia, and Y-haplogroup N in north-eastern Europe, where it is widespread today, arrived there after the Bronze Age, ca. 500 calBCE, as we detect neither in our Bronze Age samples from Lithuania and Latvia. As Uralic speaking populations of the Volga-Ural region[36] show high frequencies of haplogroup N[36], a connection was proposed with the spread of Uralic language speakers from the east that contributed to the male gene pool of Eastern Baltic populations and left linguistic descendants in the Finno-Ugric languages Finnish and Estonian[46,47]. A potential future direction of research is the identification of the proximate population that contributed to the arrival of this eastern ancestry into Northern Europe.

## Methods

**Sampling and DNA extraction.** DNA was extracted[48] from a total of 81 ancient human samples (teeth and bones) from the Eastern Baltic region, ranging from the Mesolithic Kunda culture to the Late Bronze Age (Supplementary Table 8). From Scandinavia (Sweden), we sampled 22 human remains from Mesolithic, early TRB and LN contexts. Two samples from north-western Russia were associated with Mesolithic contexts. The samples and their archaeological context are described in Supplementary Note 1 and presented in a tabular overview with sequencing results in Supplementary Data 1.

Sampling was performed in the cleanroom facilities at the Institute for Archeological Sciences in Tübingen for the Eastern Baltic material, at the Australian Centre for Ancient DNA for the Popovo sample, at the cleanroom facilities of the Max Planck Institute for the Science of Human History, Jena, for the samples from Olsund and Uzhni/Yuzhny Oleni Ostrov, and in the ancient DNA laboratory of the Archaeological Research Laboratory, Stockholm, for the remaining Swedish material. The human remains were treated with ultraviolet (UV) light from all sides for 10 min to reduce surface DNA contamination. Teeth were sawed transversally at the border of root and crown before sampling dentine powder from the inside of the crown with a sterile dentistry drill. Bone powder was taken from the inner parts of the bones with a sterile dentistry drill after removing the surface layer of the bone.

Between 30 and 200 mg of powder was used for each DNA extraction (Supplementary Data 1, column M1). The extraction was performed following a silica-column-based protocol optimized for the recovery of small ancient DNA molecules[48] with use of the High Pure Viral Nucleic Acid Large Volume Kit (Roche). Extraction buffer (0.45 M EDTA, pH 8.0 (Life Technologies), 0.25 mg/ml Proteinase K (Sigma-Aldrich)) was added to the bone powder aliquot, and rotated overnight at 37 °C. The powder was then pelleted by centrifugation at 14,000 rpm. The supernatant was added to 10 ml binding buffer (5 M GuHCl (Sigma-Aldrich), 40% Isopropanol (Merck)) with 400 µg sodium acetate, pH 5.5 (Sigma-Aldrich) and mixed. The mixture was then transferred to the High Pure Extender Assembly funnel with a purification column attached and contained in a 50-ml Falcon tube. The tube was then spun at 1500 rpm for at least 8 min with slow acceleration until the binding buffer had mostly passed the purification column. Then the column was transferred into a new collection tube and the liquid remaining in the funnel was transferred to the column that was then centrifuged at 14,000 rpm. This was followed by a wash step consisting of adding 450 µl of wash buffer (supplied with the High Pure Viral Nucleic Acid Large Volume Kit) to the column and spinning it at 8000 rpm for 1 min, the wash step is repeated and then followed by two dry spins at 14,000 rpm for 1 min. The DNA was eluted in a fresh siliconized Eppendorf tube in two elution steps of 50 µl TET (1 mM EDTA, 10 mM Tris-HCl, pH 8.0 (AppliChem), 0.05% Tween-20 (Sigma-Aldrich)) centrifuged for 1 min at 14,000 rpm, resulting in 100 µl of DNA extract for each sample. Negative controls were taken along for each extraction setup.

**Library preparation and targeted enrichment of human mtDNA.** Double-stranded next-generation sequencing libraries were prepared from a 20-µl aliquot of extract following a protocol established for ancient DNA[49]. Negative controls were taken along for each library preparation setup. First, a blunt-ending step was performed by adding the template to a mix of 1× NEB buffer 2 (NEB), 100 µM dNTP mix (Thermo Scientific), 0.8 mg/ml BSA (NEB), 0.4 U/µl T4 Polynucleotide Kinase (NEB), 0.024 U/µl T4 Polymerase (NEB) and 1 mM ATP (NEB) and incubating at 15 °C for 15 min, then for 15 min at 25 °C, followed by purification with the MinElute kit (QIAGEN) and elution in 18 µl of TET. The following adapter ligation was performed by adding 1× Quick Ligase Buffer (NEB), 250 nM Illumina Adapters (Sigma-Aldrich) and 0.125 U/µl Quick Ligase (NEB) for a final reaction volume of 40 µl. The mix was incubated for 20 min at room temperature after which another MinElute purification was performed and the DNA is eluted in 20 µl TET. The following fill-in step consisted of adding the 20 µl DNA to 1×

Isothermal Buffer (NEB), 125 nM dNTP mix (Thermo Scientific) and 0.4 U/μl Bst Polymerase 2.0 (NEB) for a final reaction volume of 40. The mix was incubated for 20 min at 37 °C and for 20 min at 80 °C. After the fill-in step, libraries were quantified via qPCR to ensure that the reactions were efficient. Some DNA extracts showed evidence of inhibition of enzymatic reactions, possibly due to the presence of humic acids or chemicals (glue or hardener) used for bone treatment[50]. To overcome inhibition, library preparation was repeated for samples that had a low DNA yield after initial library preparation or had abnormal extracts (e.g. dark colouring, floating particles, etc.) using 10-fold less extract as template to dilute the potential inhibiting factors.

Libraries were then barcoded in a PCR-reaction using primers containing sample-specific index sequence combinations[51] and limiting the amount of template molecules to 2e+10 per reaction (0.2 mM of the library-specific P5 and P7 primers, 1× Buffer Pfu Turbo (Agilent), 0.25 mM dNTP mix, 0.3 mg/μl BSA, 0.025 U/μl Pfu Turbo (Agilent) for a total reaction volume of 100 μl). The amplification took place in a modern DNA lab with an initial denaturation of 2 min at 95 °C, then 10 cycles of: 30 s at 95 °C, 30 s at 58 °C, 1 min at 72 °C; followed by elongation for 10 min at 72 °C.

Libraries were enriched for human mitochondrial DNA using a bead-based hybridization protocol[52], pooling at most five different sample libraries into one capture pool.

**Sequencing for screening.** Libraries and mtDNA-enriched library pools were quantified on an Agilent 2100 Bioanalyzer DNA 1000 chip and pooled at equimolar concentrations. Libraries not enriched for human mtDNA were shotgun sequenced to determine the percentage of endogenous human DNA in every DNA library and assign the genetic sex of the individuals[53]. Libraries enriched for mtDNA were sequenced separately to allow for reconstruction of the mitochondrial genome of each individual and estimation of modern mitochondrial contamination. Library pools were sequenced according to the manufacturer's protocols on an Illumina HiSeq2500 at the department of Medical Genetics at the University of Tübingen for 2 × 100 cycles to a depth of ~1.5 million reads per sample.

**Data processing for screening.** After demultiplexing, resulting sequencing reads were processed using a computational pipeline developed for aDNA[54] that merges paired-end reads (default parameters) and mapping of reads against a user-specified reference genome. Between 326 and 10,039,616 shotgun sequenced reads (Supplementary Data 1, column N) went into mapping with BWA (v0.6.1)[55] against UCSC genome browser's human genome reference GRCh37/hg19. For mtDNA capture, data between 375 and 7,454,704 reads (Supplementary Data 1, column Q) went into mapping against the human mtDNA reference rCRS[56] using the circular mapper implemented in the pipeline[54]. The low number of reads mapping for Spiginas3 and Motala313 indicated a failure of reagents during library preparation.

The proportion of endogenous human DNA in shotgun sequencing ranged from 0.00% to 59.6% (Supplementary Data 1, column O). Genetic sex could confidently be determined for 55 individuals[53] (Supplementary Data 1, column U).

The mtDNA reconstruction and contamination estimation was performed by an iterative likelihood-based approach, taking into account that the consensus mtDNA sequence should be reconstructed from molecules that originate from a single individual and that show characteristics of aDNA[58]. Complete mitochondrial genomes (covered at least 85%) could be reconstructed for 61 individuals and less than 5% mitochondrial contamination (Supplementary Data 1, column S). For these, the percentage of deamination at the molecule ends exceeded 20%, a characteristic of authentic ancient DNA[58] (Supplementary Data 1, column T).

The three extracts produced for the sample Olsund did not undergo the screening procedure; the mtDNA haplogroup and mtDNA contamination reported for this sample was determined from the nuclear capture data, see below.

**Nuclear capture and sequencing for genome-wide data.** Forty-one samples (including two previously studied north-western Russian samples[25]) were chosen for nuclear capture or deep shotgun sequencing. Uracil–DNA–glycosylase treated (UDG-half) libraries[59] were prepared out of the DNA extracts of these samples by adding the extract to a reaction of total volume 60 μl with 1× Buffer Tango (Thermo Fisher Scientific), 100 μM dNTPs, 1 mM ATP and 0.06 U/μl USER enzyme (NEB) and incubating for 30 min at 37 °C. The reaction was then inhibited by adding 0.12 U/μl UGI (NEB).

These libraries were then barcoded with sample-specific index sequence combinations[60], subsequently amplified with Herculase II Fusion (Agilent) and enriched using an in-solution hybridization protocol[24] for a targeted set of ~1.2 million nuclear SNPs (1240k SNP set)[2,4]. 

Enriched libraries from the Eastern Baltic and Swedish samples were paired-end sequenced on a NextSeq500 at the department of Medical Genetics at the University of Tübingen using 2 × 75 bp cycles and a HiSeq4000 at the IKMB in Kiel, using 2 × 150 bp cycles, and single-end sequenced on a HiSeq4000 for 75 cycles at the Max Planck Institute for the Science of Human History in Jena. The UDG-treated library of UzOO77 was processed at the Max Planck Institute for the Science of Human History in Jena, Germany, and was sequenced there on a

HiSeq4000 for 2 × 75 cycles, and the UDG-treated library for Popovo2 was processed at Harvard Medical School, Boston, USA, and sequenced here on a NextSeq500 for 2 × 75 cycles.

Additionally, the non-UDG-treated screening library of Gyvakarai1 was paired-end sequenced on two lanes of a HiSeq4000 for 2 × 75 cycles, and on a full run of a NextSeq500 for 2 × 75 cycles. The screening library for Kunila2 was paired-end sequenced deeper on 80% of one lane of a HiSeq4000 for 2 × 100 cycles. Additionally, 40 μl of DNA extract of Kunila2 was converted into a UDG-treated library, and pair-end sequenced on one lane of a HiSeq4000 for 2 × 75 cycles. The three UDG-half libraries for Olsund were single-end sequenced on a HiSeq4000 for 75 cycles.

Furthermore, DNA was extracted from the dense petrous portion of individual MotalaAA and converted into a UDG-half library which was shotgun single-end sequenced on a HiSeq4000 for 75 cycles. Sequencing strategies and facilities are summarized in Supplementary Data 1, column AE.

After demultiplexing, resulting sequence data were further processed using EAGER[54]. This included mapping with BWA (v0.6.1)[55] against UCSC genome browser's human genome reference GRCh37/hg19, and removing duplicate reads with same orientation and start and end positions. To avoid an excess of remaining C-to-T and G-to-A transitions at the ends of the reads, two bases of the ends of each read were clipped for each sample except for the non-UDG-treated data for Gyvakarai1, where 10 bases from each end were clipped.

For each of the targeted 1240k SNP positions, a read was chosen at random to represent this position using the genotype caller *pileupcaller* (https://github.com/stschiff/sequenceTools).

**Quality control of genome-wide data.** The samples that were covered at <10,000 SNPs of the 1240k SNP set were excluded from further analyses. We evaluated the authenticity of the samples by observing typical patterns of deamination towards read ends (Supplementary Data 1), estimating heterozygosity on the mtDNA with schmutzi[58] and heterozygosity on the X chromosome in male samples with ANGSD[61] (Supplementary Data 1), and evaluating the ratio of the reads mapping to X and Y which showed no outliers (see below).

We observe that all our individuals predating the LN appear genetically distinct from any modern-day population that could have contaminated them, and that female samples cluster together with their male counterparts from the same archaeological cultures (Fig. 2), which gives indirect support to the authenticity of our data.

We excluded Saxtorp5158 from our analysis due to its high degree of contamination on the mtDNA, and Saxtorp389 as it showed unusual ancestry for its dating and archaeological context, consistent with modern European contamination.

Using the software READ[62], we determined individuals Kretuonas2 and Kretuonas5 to be identical twins, consistent with a value of >0.5 for $f_3$(Kretuonas2, Kretuonas5; Mbuti). We do not include the lower coverage sample Kretuonas5 in ADMIXTURE analysis and other analyses that cluster the individuals into one population, thereby mitigating bias resulting from a defined population consisting of closely related individuals.

We merged our data of Kunila2 with previously published data of the same individual[22] after confirming the identity with outgroup $f_3$ and READ.

**Sex assignment.** Genetic sex of the 41 selected samples was assigned using shotgun and SNP capture data by calculating the ratio of average X chromosomal and Y chromosomal coverage to average autosomal coverage at the targeted SNPs (X and Y rate, respectively). Samples with an X rate between 0.65 and 1 and a Y rate between 0 and 0.15 were assigned female and those with an X rate between 0.35 and 0.55 and a Y rate between 0.4 and 0.7 were assigned male, supporting the informative value of the Y rate over the X rate using this method (Supplementary Fig. 9), as demonstrated previously[5].

**Population genetic analyses.** Due to the nature of ancient DNA research, no predetermination of sample size by statistical methods was carried out and there was no randomization of experiments or blinding of investigators to allocation during experiments and outcome assessment.

Reference datasets for ancient populations are taken from the publicly available dataset used in refs. [5,6,21,22,] (which includes genotypes from samples published earlier in refs. [2–4,9,64], among others), as well as genotyping data from worldwide modern populations (Human Origins or HO dataset) published in the same publications and provided by the David Reich lab[6]. When analyzing only ancient samples, we make use of the 1,196,358 SNPs targeted by the 1240k SNP capture, using the genotypes of deep shotgun sequenced modern Mbuti as the outgroup. For analyses involving modern populations, we restrict to the intersection of 597,503 SNPs between the 1240k SNP set and the HO dataset.

PCA was performed with *smartpca* in the EIGENSOFT package[64] by constructing the eigenvectors from modern West Eurasian populations (Supplementary Fig. 7) and projecting ancient individuals on these eigenvectors (Fig. 2a, Supplementary Fig. 8).

Admixture analysis (Fig. 2b) was carried out with ADMIXTURE on 3784 modern and 378 ancient individuals for ancestral clusters $k = 2$ to $k = 16$ with 100 bootstrap replicates. The SNP dataset was pruned for linkage disequilibrium with

PLINK using the parameters --indep-pairwise 200 25 0.5. We considered the cross-validation (CV) error and report in Fig. 2b and Supplementary Fig. 4 the results of $k = 11$, where the CV error levels out at a minimum.

To quantify population affinities and admixtures suggested in the PCA and ADMIXTURE analysis, we carried out $f$-statistics using the programs $qp3Pop$ and $qpDstat$ in the ADMIXTOOLS suite (https://github.com/DReichLab) for $f_3$- and $f_4$-statistics, respectively. $f_3$-statistics of the form $f_3(X,Y; Outgroup)$ measure the amount of shared genetic drift of populations X and Y after their divergence from an outgroup. Admixture $f_3$-statistics of the form $f_3(Test;X,Y)$ indicate when significantly negative that population $Test$ is intermediate in allele frequencies between populations X and Y and could be considered an admixed population. This test was performed with parameter $inbreeding:YES$ and cannot be done for populations with less than two individuals. $D$-statistics of the form $D(X,Y; Test, Outgroup)$ show if population $Test$ is symmetrically related to X and Y or shares an excess of alleles with either of the two. Results are only reported for statistics based on more than 10,000 SNPs.

To formally test for the number of source populations and the proportion of ancestry these contributed to our studied populations, we used the $qpWave$ and $qpAdm$ programs from ADMIXTOOLS. These programs implement the methodology of using regression of $f_4$-statistics of a $Reference$ population with various outgroups to relate its ancestry to a $Test$ population[2,6]. With $qpWave$, we identified potential source populations for our population under study by testing if a set of $Left$ populations (the $Test$ population under study and its potential proximate source $Reference$ populations) is consistent with being descended from $n$ waves of admixture which have unique relationships to the $Right$ outgroup populations (Mbuti, Papuan, Onge, Han, Karitiana, Mota, Ust Ishim, MA1, Villabruna). This is given when rank $n-1$ cannot be rejected ($p > 0.05$), and rejected (i.e. more than $n$ waves of admixture are needed to explain the ancestry of $Test$ and $Reference$) if rank $n-1$ can be rejected ($p < 0.05$).

To estimate admixture proportions, we used $qpAdm$ to model the Test population as a mixture of various source populations postulated from the $qpWave$ test, setting as $Left$ populations the $Test$ and source populations and as the $Right$ populations the various outgroups named above.

**Data availability**. The sequence data reported in this paper are deposited in the Sequence Read Archive (Accession numbers: SAMN08139261–SAMN08139301) and complete mitochondrial consensus sequences are deposited in GenBank (Accession numbers MG428993–MG429049).

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

## Acknowledgements

We thank Isil Kucukkalipci, Cäcilia Freund, Antje Wissgott, Guido Brandt, Nadin Rohland and Swapan Mallick for technical support in DNA analyses, and Herve Bocherens, Vesa Palonen, Anne-Maija Forss and Igor Shevchuk for arranging sample treatment and technical support for radiocarbon analyses. We thank Ben Krause-Kyora for use of the sequencing facilities at the IKMB (Kiel). We thank Henny Piezonka, Cosimo Posth and Iosif Lazaridis for helpful discussion and suggestions. F.H. thanks Maj-Lis Nilsson and Erika Rosengren for support regarding Saxtorp site, and Elise Hovanta, Anna Larsdotter and Kristina Lindkvist for help regarding the Ölsund site. C.-C.W was supported by the Max Planck Society and Nanqiang Outstanding Young Talents Program of Xiamen University. J.K. and A.M. were funded by DFG grant KR 4015/1-1 and the Max Planck Society. E.B. was funded by the RFBR grant 16-06-00303. D.R. is supported by US National Science Foundation HOMINID grant BCS-1032255, US National Institutes of Health grant GM100233, an Allen Discovery Center grant from the Paul Allen Foundation, and is an investigator of the Howard Hughes Medical Institute.

## Author contributions

A.M., R.J. and J.K. conceived the idea for the study. M.D., G.Z., F.H., R.A., V.M., V.K., A. V., E.B. and R.J. assembled the skeletal material. A.M., S.P., A.F., A.A.V., M.F., C.E., M. O., D.R., W.H. and J.K. performed or supervised the wet lab work. A.M., C.-C.W. and S. P. analyzed the data. A.M., C.-C.W., M.D., G.Z., F.H., M.T., R.A., M.O., W.H., S.S. and J. K. wrote the manuscript and supplements.

## Additional information

**Competing interests:** The authors declare no competing financial interests.

