## [Peer Review File · Nature Communications]

Reviewers' comments:

Reviewer #1 (Remarks to the Author):

In this paper, Alissa Mittnik and colleagues present new genomic informations of 24 additional ancient Northern European samples from a time period spanning more than 7,000 years to the growing knowledge on prehistoric genomic variation. For two of the samples, whole genome data was obtained with deep shotgun sequencing, and the remaining 22 samples were sequenced using nuclear target enrichments of pre-selected targeted SNPs. In the manuscript, the genetic affinities of these samples to each other and to other ancient as well as modern samples are quantified, which lead to the conclusion of population continuity in hunter-gatherer ancestry from the Mesolithic to the Neolithic in the Baltic, the ancestral migration of farmers from Central Europe to Scandinavia early on, and a better characterization of the genetic contribution of the "Steppe" migration to the genetic variation of the region.

The article is generally well written, and care has been taken to qualify results, as opposed to the highlighted summary frequency seen in other papers. This is definitively appreciated, but does render some paragraphs hard to follow. All analysis presented (D-statistics, PCA and ADMIXTURE) were conducted rigorously and to a high quality standard. The archaeological information, neglected in some articles in the ancient DNA field, is well summarized and presented, as far as I can judge, especially in the supplementary information.

While the relatively large number of 24 additional ancient samples is definitively a strong point of the paper, the paper seems to lack a focus on actual research questions. This is a major drawback that persist throughout the paper. The introduction, for instance, is nicely summarizing our current archaeological understanding, but fails to discuss what questions remain open and how they should be addressed. Similarly, the result sections are very descriptive and the lack of questions makes them a bit hard to follow.

The article also lacks any major contribution that was not previously presented from existing samples. While confirmation is important to science, I wonder whether the paper would not fit better to a more specialized journal, especially since there is also no methodological advancement presented, despite the fact the at least the higher quality genomes would have allowed for more advanced analyses (e.g. making use of genotyping information through genotype likelihoods, rather than resorting to standard tools using random allele sampling).

Indeed, while the article expands over previously published information from each period studied, it fails to advance previously obtained knowledge and will therefore be of interest mostly to a specialized audience interested in regional differences on a small scale and to researchers utilising the data for other purposes. Let me be specific:

- While the authors demonstrate that the hunter-gatherer samples from the western and eastern Baltic region have rather distinct ancestries, this was shown previously (e.g. Haak et al. 2015, Lazaridis et al. 2016 and Jones et al. 2017), even if on a larger scale.
- The genetic similarity (and potential continuity) between Mesolithic and Neolithic (pottery-wielding) hunter-gatherers was demonstrated in a (unfortunately) very recent article (Jones et al. 2017), but with fewer samples.
- The genetic turn-over consistent with the change of subsistence in Scandinavia was previously demonstrated (e.g., Skoglund et al. 2012 and Skoglund et al. 2014). While the samples presented in

this manuscript indeed come from a slightly earlier chronological period, they stem from the same culture (TRB), and the finding of gene flow from Central Europe presented in this manuscript was not unexpected.

- The major genetic turnover in Europe in the Late Neolithic/Early Bronze was previously shown also for Baltic samples (e.g., Allentoft et al. 2015), though it is novel that some of the individuals show "Steppe-like" ancestry independent of Corded Ware culture.

- Finally, the finding of a resurgence of hunter-gatherer ancestry in later periods was well described before (e.g. Haak et al. 2015). Again, though, this study does add regional detail by demonstrating this resurgence to have happened later in the Baltics than Central Europe.

In summary, while the paper presents a lot of additional regional details, I feel it lacks findings of broader interest worthy of a publication in Nature Communications.

In addition, I also have some methodological criticisms, as detailed below. However, I'm confident that the authors could address all of these in a future revision.

1) The approach to use only reads exhibiting post-mortem damage patterns to assess contamination through D-statistics is interesting. However, there are some issues with this approach. Firstly, the lack of a gene-flow signal does not necessarily imply absence of contamination. This is especially true if the test populations chosen are unlikely sources for the contamination (e.g. diverged Asian and African populations). To test for inter-sample contamination, the other samples should probably be used, or at least modern and ancient populations from the region (e.g. contemporary populations of Scandinavian, Baltic and German origin, or the samples of Skoglund et al. 2014). Secondly, PMD patterns can easily result from other sources (e.g. UV light, inter-sample contamination), hence comparing D-statistics of PMD affected reads and other reads is likely underpowered, especially if there are only few reads. Thirdly, it is unclear how many reads are actually required to detect contamination using this approach, and I doubt that fewer reads are required than for the usually used methods. In summary, I think this approach could be an interesting methodological addition to the paper, but the authors would need to show with some artificial (bioinformatic) contamination from both other samples as well as modern Baltic sources that the amount of data they have was sufficient to detect it.

2) The lack of contamination estimation of the Popovo2 sample (it had too few reads for ANGSD contamination estimation, and is also missing mtDNA contamination information) makes it hard to consider this sample as uncontaminated. Given the expected abundance of mtDNA reads (e.g. off-target reads), a direct mtDNA contamination estimate should at least be attempted. As mentioned above, the D-statistics method is not sufficient unless its power is well demonstrated. Without demonstrating that this sample is uncontaminated, it must be removed from the analysis.

3) There is also a major issue with the highly contaminated sample Saxtorp5158T (50-52% mtDNA contamination according to Extended Data 1). This sample was found to be related to another sample from the dataset and excluded from any further analysis because of their first-degree familial relationship. I emphasize that this sample should have been excluded on the basis of the high contamination estimate alone. Unless further information about the nature of the contamination is provided (e.g. what and how many different contaminating haplogroups; software schmutzi by Renaud et al. 2015 could be of use, maybe it was even used for the original mtDNA contamination estimates), it cannot be excluded that even the first-degree relationship between the samples could have resulted from an inter-sample contamination (undetectable via PMD filtering applied here to correct for the contamination). I therefore do not find the inference of the familial relationships of this sample valid

and I suggest it is excluded from the paper. The correct reason for the exclusion of this sample is important because familial relationships could be used in interpretations of archaeological contexts.

4) The method used to “call” SNPs was to randomly select a single read at each position. This can hardly be termed “calling genotypes”, and the authors should rephrase that statement. Given the often relatively high coverage, especially on targeted SNPs (up to 8.8x), it is disappointing that only a very small portion of the original information was actually used. In recent years, it become more common practice to at least select the most frequent allele to represent the position, which decreases the error rate as a function of coverage. More recently, tools have also been published that use the information from all the reads probabilistically and take into account both sequencing error rate and PMD, to call both genotypes as well as pseudo-haploids (e.g., Hofmanova et al. 2016, Link et al. 2017), and I suggest to the authors to consider using such methods in future. This is particularly relevant for studies like this one in which very different sequencing platforms and centers were used. Samples thus likely differ in their sequencing errors (both in amounts as well as type) and post-mortem damage profiles, and by sampling an allele randomly, this results in genetic distances hard to compare. While I do not request that the authors redo the complete analysis using a more appropriate calling method (especially since their results are consistent with previously published results), they should at least add to the Extended Data 1 all information regarding where and on which platform each sample was sequenced, so readers concerned with this issue might judge the consistency of the inferences and potential bias from created by the different sequencing strategies.

5) The authors mention from the line 263 onwards that “We demonstrate here that early Swedish farmers, similarly, derive from a mix between Early Farmers and WHG, without additional gene-flow from foraging groups represented by SHG from Central Sweden. However, as no forager samples from South Sweden, Denmark or Northern Germany have been studied as of yet we cannot say if the genetic substrate was more WHG-like in these regions and if local admixture might have played a greater role in the TRB culture.” They also mention that they have incorporated in their analysis samples from Fu et al. 2016 (line 522) where there are among others Mesolithic WHG samples from Berry au Bac and Chaudardes (Northeast France) and Bockstein and Ofnet (from Germany). Also a sample from Loschbour, Luxembourg is available (Lazaridis et al. 2014). It is hard to assume that these less than 10,000 years old hunter-gatherers could not be very advantageously used to represent the genetic variability of WHG in these regions. It would therefore significantly add to the paper if such samples were utilised to investigate if the observed hunter-gatherer admixture in TRB samples was of a local Baltic origin or not.

6) The observed relationships of Mesolithic hunter-gatherers (EHG-SHG-Kunda-WHG) appear consistent with an East-West and North-South pattern of isolation by distance. Such a scenario is further supported by relatively small and step-wise differences between the samples in the ADMIXTURE analysis. The main support for these hunter-gatherer populations being distinct seems to come from the PCA. However, since all ancient DNA samples were projected on genetic variation of modern West Eurasians, the position is not very accurate and small differences between ancient samples should not be overinterpreted, especially since the samples locations are rather far apart geographically. In addition, the later Narva samples appear intermediate between the SHG and WHG groups, further questions this clear distinction. It might thus be better if samples were analysed on an individual basis where it is possible (D-statistics, qpWave), rather than pooling based on PCA coordinates.

7) On the lines 237-242, the authors write “Notably, in addition to haplogroup H, the maternal lineages seen in eastern Baltic samples (n=31; Supplementary Information Figure S5) encompass all of the major haplogroups identified in complete mtDNA genomes from Holocene Scandinavian and western European hunter-gatherers (n=21:U2, U5a, U5b) 12 , as well as haplogroup U4 which has

been found in high frequency in Mesolithic foragers from Russia 24 and K1, a derivative of the U8 branch found in Scandinavian foragers 20, suggesting a large census size of Baltic hunter-gatherers to maintain such genetic diversity". The statement on the increased population size is also present in the Conclusion (the line 418). Comparing genetic diversity based on haplogroup distributions is tricky, since haplogroups are not genetically equally distant from each other. In addition, the arbitrary pooling of haplogroups to "major haplogroups" has a huge impact on such a comparison. However, genetic diversity can be easily calculated directly from mitochondrial sequences without using haplogroups assignments. To make a statement regarding increased diversity then requires a proper comparison to a hunter-gatherer sample encompassing a geographical region of similar size and a similar chronological transect (e.g. selection of data from Brandt et al. 2013 and Post et al. 2016).

8) Starting from line 242, the authors write "We see in Baltic foragers no genomic evidence of gene-flow from Central European farmers or any Y-chromosomal or mitochondrial haplogroups that are typical for them, suggesting that any traces of agriculture and animal husbandry in the Baltic Early and Middle Neolithic were due to local development or cultural diffusion." Yet starting from the line 232: "The Narva individual Spiginas1 (dated to ca. 4440–4240 cal BCE) belongs to a mitochondrial haplogroup of the H branch providing the first direct evidence that this branch was present among European foragers without gene-flow from farmers (Table 1, Figure 2b)." and "... and K1, a derivative of the U8 branch found in Scandinavian foragers" on line 240. Using the argument of the mitochondrial haplogroup in the first statement is difficult since, as the authors mention in the second statement, some of the found haplogroups are actually not typical for hunter-gatherers (K1 and H). Indeed, in the reference (20) given, which is (Malmström et al. 2015), the haplogroup was used as an argument in favor of admixture with farmers. Interestingly, the only argument for K1 being found in hunter-gatherers of the region is found their own manuscript, namely in sample Donkalis5, which is dated very early (6075–5920 calBCE). Since this is an interesting result, also to support some of the other claims, it would be interesting to discuss the archeological interpretation for this sample. In doing so, please note that K1 has so far only been reported in pre-neolithic samples from Greece.

9) Admixture was performed (as sometimes seen) at the same run with both modern and ancient populations. The definition of ancestral clusters in the analysis is therefore hindered by the heterochronous character of the data. Since the authors interpret also only very small ancestry components and differences in these between samples, it would be important 1) to check that these shared components are not artifacts of running the analysis with a huge panel of distantly related modern samples 2) to add existing ancient samples that would help in the made interpretation. For instance, to interpret the component maximised in Native Americans as a evidence for ancestry related to the MA1 and AG3 samples, these two ancient samples must be included in the analysis (I note that I did not have Extended Data Figure 4, maybe the samples were included there?). Also, bootstrapping should be used to add confidence intervals and to test whether these small components are actually meaningful.

Minor points:

- Extended Data Figures 1-4 were missing from the information provided. Also, Extended Data Figures 1-3 are never cited in the main text.

- Mitochondrial contamination is mentioned in the Quality control section of the methods for sample Saxtorp5158, and the mtDNA contamination values are presented in the Extended Data 1 for other samples. However, the method used to obtain these estimates was nowhere described.

- The authors should avoid calling the nuclear capture samples "genomes" to prevent confusion (e.g. line 135). Since these are capture data, it is only possible to refer to those as having "genome-wide

data", as is commonly done for modern captures.

- The title "The genetic history of Northern Europe" is not very descriptive and informative about the content of the article. While it may be difficult to pinpoint an obvious major finding for a title, it could be bound at least chronologically.

- On the line 165, Fig. 2a not 2A should be used. Also, for consistency only Fig. 2a (abbreviated, e.g. line 165) or Figure 2a (not abbreviated, e.g. line 532) should be used.

- On the line 174, "its geographical vicinity" should be "its geographically vicinity".

- The coverage for the two whole-genome shotgun samples (Kunila2 and Gyvakarai1) was only reported for the targeted SNPs. It would be useful to report also genome-wide coverage of these samples, especially since a full genome with over 7x coverage is one of the highlights of the paper and valuable by itself.

Reviewer #2 (Remarks to the Author):

The paper focuses on the genetic history of Northern Europe, a key region for which there is only a limited amount of paleogenomic data. The paper reports genome-wide DNA data from 24 ancient North Europeans spanning the transition to agriculture, and the Bronze Age.

The paper's main claims are that (1) Scandinavia was settled after the retreat of the glacial ice sheets from a southern and a northern route, (2) the first Scandinavian Neolithic farmers derive their ancestry from the Anatolian farming dispersal into Europe, 1000 years earlier than has been previously reported, (3) the range of Western European Mesolithic hunter-gatherers (WHG) extended to the east of the Baltic Sea, and persisted without gene flow till around 2,900 BC, the timing of the arrival of steppe populations (Corded Ware Culture) to this region.

The manuscript is well written and easy to follow, and the figures and tables are of excellent quality.

The claims are novel and provide an important contribution to the study of the genetics of prehistoric Europeans and this paper is a valuable contribution to the field. The analytical approaches applied are excellent, and convincing (including the correct use of relevant published ancient and modern genomic data in the various analyses).

The following points need to be addressed, as I think that they will improve the manuscript and clarify a few elements.

1. Line 208: "An eastern Baltic refugium of European hunter-gatherers" . I am not convinced that the paper provides any support for a Baltic refugium and that Scandinavia was settled after the retreat of the Ice Sheets. This fact is well reported from archaeological and paleoclimatic data, but the data presented here include previously published EHG genomes (Oleni Ostrov and Popovo) as well as new Kunda Culture Mesolithic genomes from Lithuania> the latter are dated to the 5th millennia BC, which is several millennia after the retreat of the Ice Sheets. As such, the results of the genetics of these individuals do not provide any direct support for the timing and nature of the colonization of Scandinavia. Perhaps the authors refer to genetic continuity between hunters and farmers but this is another story as the term 'refugium' is held for regions that were occupied during the LGM (and were not necessarily ice free).

2. Line 115" Here we genetically investigate.." I suggest that the authors will provide here some

concrete research questions. Without these the reader is left with the feeling that the paper just explores dynamics of populations and their admixture/turnovers, but in fact I find this paper to be well structured and clear in the specific aspects which it focuses on (as is evident in the results section from the systematic tests applied). I think that this can strengthen the paper and make it easier to follow.

3. Results and Discussion section, Samples and archaeological background: there are major chronological gaps between the samples. This is fine, as this is the nature of the field, but it is important to mention this as some archaeological phases are not well represented.

4. Lines 152-4: "...these samples address the open question whether the first introduction of farming around 4,000 BCE was driven by newcomers or by local groups involving later gene-flow from Central European farmers". This should be discussed above, perhaps when outlining the main questions addressed.

5. Line 248 "Early Neolithic Migration into Scandinavia". This title is confusing. If it is Early Neolithic then why does the D statistic is comparing TRB to Middle Neolithic Central Europe, X, Mbuti? Should the MN C Europe not be replaced by an Early Neolithic either LBK or Anatolian? It is logical to show that there is no significant results and that we can assume that TRB is closely affiliated with the MN Central Europe, but then this does not imply directly that TRB (the first introduction of agriculture to South Sweden) was the outcome of the demic diffusion. If this is the case, then it is necessary to at least compare it to LBK as MN are not early farmers and already further admixed with WHG than the first farmers. The same argument applies for eth PWC, should it not be compared to early Neolithic instead of MN? In any case there is no evidence for an actual demic diffusion. It is best to discuss the appearance of the first farmers in Scandinavia but since there is a major time delay between this event and the first demic diffusion into Europe, I think that it is better not to apply this term (also since there is no evidence for an actual logistic population growth and gradual diffusion and this is not tested here).

6. Line 518: Population genetics analysis. I think that there is a sentence missing about the D statistics.

7. Line 542: should be "...population X and Y after..."

8. Supplementary Information Table S4. Significantly positive results for D(Ancient Eastern Baltic population, A; X, Mbuti). I think that the authors need to revise the description below it as it discusses 'modern' rather than 'ancient'.

Reviewer #3 (Remarks to the Author):

- Mittnik et al. analyse key samples from the Baltic region spanning the Mesolithic, Neolithic and Bronze age. This is a nice addition to the growing literature on ancient European genomics. Although perhaps a bit lengthy, the manuscript is well written and provides genetic support for two independent movements into Scandinavia. There are specific concerns regarding the heavy reliance on qpWave and qpAdm, and the exact interpretations of the results of these analyses. However, most conclusions can be reached without the need of these analyses. Publication is supported, perhaps with some shortening, provided the qpWave/qpAdm issues are addressed. More specific comments follow.

- There appears to be some confusion regarding what qpWave and qpAdm are actually able to test. The link between D-statistics and qpWave is explicit in that qpWave uses a matrix of F₄s, thus I believe many of the qpWave/qpAdm conclusions could be reached from D-stats alone (additional tests not shown in the manuscript will be required in some cases). D-stats are now widely understood, while qpWave and qpAdm are not. The interpretability of D-stats is also simpler, as it does not require an understanding of linear algebra, potentially broadening the readership. I strongly recommend (1) reframing the discussions by focussing on results in the D-stats, and (2) any remaining qpAdm tests (to show mixing proportions) must be justified (see below).

-

- Define calBCE in Abstract, or avoid. Describe EDAR allele function.

-

line 139: 7,500 to 5,500 calBCE for the two Mesolithic Russians is inconsistent with Tables 1 and S1. Tables 1 and S1 suggest 5500-5000 BCE for Uz0077 and Popovo2. SI1 has Uz0077 dated indirectly from associated individuals to 7450-6950 calBP and Popovo2 indirectly as 9500-7500 BP or maybe younger.

- line 174 grammar: 'its geographically vicinity'.

- line 185-187: My understanding is that qpWave gives a lower bound on the number of distinct ancestry components in one group (left populations) that are derived from another group (right populations). It is not clear why qpWave would be able to reject the hypothesis that SHG is formed by admixture of WHG and EHG. The qpAdm test for SHG as a linear combination of WHG and EHG gives admixture coefficients as quoted from Table S9, and this cannot be rejected. I guess the text of line 185 should say qpAdm, not qpWave? However, the validity of performing this qpAdm test has not been established by first showing that qpWave with left populations={WHG, EHG} has matrix with full rank (rank=1, i.e. two source populations). I did not see such an entry in Tables S6/S7. I note that the other criterion for this qpAdm test is that qpWave with left populations={SHG,WHG,EHG} has the same rank (i.e. rank=1), and this is confirmed in Table S7.

- lines 187-191: The conclusion that the crimson component in Fig 2a is maximized in modern Native American populations is not clear from Fig 2a, as no Native American nor MA1/AG3 samples are shown. Could we see the ADMIXTURE output that is the source?

- line 200: Some care needs to be taken with regard to wording and the results being referred to. Table S6 contains qpWave results, which does not do ancestry modelling. This point aside, one might expect that there should be an entry in Tables S6/S7 for qpWave with left populations={EHG,WHG} and also left populations={SHG,WHG} (in 'contrast' to the left population={Kunda,WHG}). As I noted earlier, the former entry is absent.

-

- line 217: '[Narva] can also be accounted for by admixture of Kunda with either EHG, SHG or WHG (SI Table S6)'. I believe this is an over interpretation of the qpWave results. You have left populations={Narva, Kunda}, and qpWave cannot reject rank 0. Then you add another group to the left populations (e.g. EHG), and find qpWave is unable to reject rank 1. But this does not mean that Narva has any ancestry from the added group. There is now an additional distinct ancestry component in the left populations (derived from the outgroup populations), regardless of whether Narva has any of this newly accounted for ancestry. Looking at the separate qpWave results for NarvaEast and NarvaWest, it is clear that the most parsimonious interpretation is that NarvaWest is directly descended from Kunda, while NarvaEast has additional ancestry related to one of EHG,WHG,SHG - further, the qpAdm results for NarvaEast indicate it must be EHG or SHG and not WHG.

line 221: Unlike for NarvaEast, the qpAdm tests for left populations = {NarvaWest, Kunda, EHG or SHG} are not valid, as qpWave could not reject rank 0 for left populations = {NarvaWest, Kunda}. I suggest that a similar interpretation may be made by contrasting D-stats of the form D(NarvaEast, Kunda, X, Mbuti) with D(NarvaWest, Kunda, X, Mbuti), where X={EHG,SHG, etc}.

- lines 254-261: D-statistics are sufficient to make the point. Having the same result from qpWave is tautological as qpWave uses an F4, which is an unnormalised D-statistic. Again, the qpAdm results are not valid as you cannot reject the more parsimonious scenario of no admixture.

-

- line 274: qpWave analysis is redundant - you already have the D-stats.

- lines 277-281: qpAdm tests are not justified. You should show D-stats of the form D(PWC,SHG,X,outgroup), where X includes more populations from your qpWave right populations.

- line 287: it is not possible to test migration directionality using any of the methods employed here. Specifically, qpAdm finds the best linear combination of source populations which accounts for the target population. It is not unthinkable that swapping the target population with one of the source

populations will also give a result where the admixture coefficients are all positive.

- lines 307-314: I certainly prefer the language 'are consistent with' here, compared with the language of earlier paragraphs. The same conclusions ought to be reached from looking at individual D-stats of the form $D(\text{Baltic_LN_*}, \text{CordedWare_Central}; X, \text{Mbuti})$, with each of the Baltic_LN_* subpopulations.

- line 324: please justify this qpAdm analysis with the relevant qpWave prerequisites.

- lines 365-375: More qpWave interpretations... The cause of Baltic LN giving higher p-values over Scandinavia LNBA is apparent from the D-stats in table S5, which suggests a Levantine influence. This does not support the conclusion that Oslund is more closely related to Baltic LN, in fact the opposite appears to be true (D-stats and PCA both indicate this). Please consider carefully the relationship between qpWave results and the relevant D-stats.

-

- line 516: should be figure S6. It is not clear from the methods if only the shotgun data was used for sex determination, or if capture data were used also. I suspect this method would work poorly (if at all) for capture data.

-

- line 542: should 'X and X' be X and Y?

- line 544: should 'A and B' be X and Y?

- Supplementary Tables S4,S5,S10,S11,S12,S13,S14 (D-stats tables): Please make sure the wording in the figure legends matches the column headers. I found this wording very confusing in determining if using the BABA-ABBA convention was used, or the ABBA-BABA convention.

- Supplementary Tables S6, S7 (qpWave tables): The column headers 'Test', 'Reference A', and 'Reference B' don't make sense in this context. Any of the left populations are mathematically interchangeable and will give the same result (likewise for the right populations).

- Supplementary Tables S8, S9 (qpAdm tables): Many of the rows in these tables correspond to tests which are either not justified by relevant qpWave results, or are shown to be unjustified by relevant qpWave results. There are two conditions that should be satisfied to justify using qpAdm, quoting from Haak et al. SI10, pg 129:

``Thus, if T is admixed, as above, pick a set of outgroup populations O, and

1. Check, setting left populations $L = S$, and right populations O that the matrix X has full rank $n-1$.

2. Check, again setting $L = \{T, S\}$ that there is no strong evidence that the rank of X increases with the addition of T. '

- Supplementary Section 3: Y haplogroup assignment has been done based largely on C-T and G-A polymorphisms, many with only 1x or 2x. These are mostly UDG half treated libraries, but as methylated cytosines deaminate to thymine it would be sensible to avoid sites in a CpG context where most methylation will occur. I appreciate that there are multiple sites contributing to haplogroup assignment, and no conclusions are drawn regarding the more specific haplogroup subtypes.

Response to Referees

Responses in cursive and marked yellow

Reviewers' comments:

Reviewer #1 (Remarks to the Author):

In this paper, Alissa Mittnik and colleagues present new genomic informations of 24 additional ancient Northern European samples from a time period spanning more than 7,000 years to the growing knowledge on prehistoric genomic variation. For two of the samples, whole genome data was obtained with deep shotgun sequencing, and the remaining 22 samples were sequenced using nuclear target enrichments of pre-selected targeted SNPs. In the manuscript, the genetic affinities of these samples to each other and to other ancient as well as modern samples are quantified, which lead to the conclusion of population continuity in hunter-gatherer ancestry from the Mesolithic to the Neolithic in the Baltic, the ancestral migration of farmers from Central Europe to Scandinavia early on, and a better characterization of the genetic contribution of the “Steppe” migration to the genetic variation of the region.

The article is generally well written, and care has been taken to qualify results, as opposed to the highlighted summary frequency seen in other papers. This is definitely appreciated, but does render some paragraphs hard to follow. All analysis presented (D-statistics, PCA and ADMIXTURE) were conducted rigorously and to a high quality standard. The archaeological information, neglected in some articles in the ancient DNA field, is well summarized and presented, as far as I can judge, especially in the supplementary information.

While the relatively large number of 24 additional ancient samples is definitely a strong point of the paper, the paper seems to lack a focus on actual research questions. This is a major drawback that persist throughout the paper. The introduction, for instance, is nicely summarizing our current archaeological understanding, but fails to discuss what questions remain open and how they should be addressed. Similarly, the result sections are very descriptive and the lack of questions makes them a bit hard to follow.

We now added explicit research questions to the introduction (please see response to Reviewer 2 below). Our revision now includes 14 additional individuals, totaling 38 individuals that we analyse. Among them are individuals from periods and regions that are not described yet, such as prehistoric samples from the southern part of the Eastern Baltic region, individuals associated with the Narva Culture from Estonia and a total of 14 individuals from the Baltic Bronze Age. As the archaeological record and chronology of northern European prehistory is so distinct to rest of Europe, possibly leading to the unique genetic composition seen in the populations inhabiting the area today, a detailed look into the genetic history is justified and necessary.

The article also lacks any major contribution that was not previously presented from existing samples. While confirmation is important to science, I wonder whether the paper would not fit better to a more specialized journal, especially since there is also no methodological advancement presented, despite the fact that at least the higher quality genomes would have allowed for more advanced analyses (e.g. making use of genotyping information through genotype likelihoods, rather than resorting to standard tools using random allele sampling).

Indeed, while the article expands over previously published information from each period studied, it fails to advance previously obtained knowledge and will therefore be of interest mostly to a specialized audience interested in regional differences on a small scale and to researchers utilising the data for other purposes. Let me be specific:

- While the authors demonstrate that the hunter-gatherer samples from the western and eastern Baltic region have rather distinct ancestries, this was shown previously (e.g. Haak et al. 2015, Lazaridis et al. 2016 and Jones et al. 2017), even if on a larger scale.

We demonstrate for the first time that the Eastern Baltic was a region of continued interaction between Eastern and Western hunter-gatherers and that there was a cultural correlation to the proportion of WHG/EHG ancestry seen in individuals.

- The genetic similarity (and potential continuity) between Mesolithic and Neolithic (pottery-wielding) hunter-gatherers was demonstrated in a (unfortunately) very recent article (Jones et al. 2017), but with fewer samples.

We double the number of individuals providing genetic data from the Eastern Baltic Mesolithic, Early and Middle Neolithic hunter-gatherers and can so not only demonstrate genetic continuity between the Mesolithic and Neolithic populations but are able to investigate whether their genetic composition is due to a long-standing genetic cline formed by isolation-by-distance or more recent gene flow. Thereby we can address the archaeological hypothesis that 'Neolithisation' in the Eastern Baltic is not brought upon by a population movement from European farmers but through changing networks to hunter-gatherer groups to the East.

- The genetic turn-over consistent with the change of subsistence in Scandinavia was previously demonstrated (e.g., Skoglund et al. 2012 and Skoglund et al. 2014). While the samples presented in this manuscript indeed come from a slightly earlier chronological period, they stem from the same culture (TRB), and the finding of gene flow from Central Europe presented in this manuscript was not unexpected.

The idea that early farming in South Scandinavia was an autochthonous development through acculturation of local hunter-gatherers is common among some archeologists, and the fact that previously studied TRB individuals dated to a later phase which coincided with substantial changes in

material culture was used as a criticism and caveat to previous genetic interpretations. Our result is therefore significant, as we demonstrate for the first time that also an individual from the early phase of TRB (the earliest dated human remains from Sweden associated with an agrarian assemblage) derived most of her ancestry from Anatolian farmers and can dispel the hypothesis that the earliest adoption of farming here was due simply to acculturation.

- The major genetic turnover in Europe in the Late Neolithic/Early Bronze was previously shown also for Baltic samples (e.g., Allentoft et al. 2015), though it is novel that some of the individuals show “Steppe-like” ancestry independent of Corded Ware culture.

While the arrival of Steppe-like ancestry in the Eastern Baltic in the Late Neolithic/Early Bronze Age was indeed shown previously, the crucial new observation is that this ancestry seems to have arrived in the early phase of the Late Neolithic without the component derived from Anatolian farmers, i.e. not in the typical mixture of components found in the Central European Corded Ware (most of which are younger than 2600 BCE). Therefore we propose that at least for the Baltic region the early phase of the Corded Ware Complex (also known as Battle Axe Culture locally) was due to a movement of a Yamnaya-like population directly from the Steppe without previous contact to Central European farmers.

- Finally, the finding of a resurgence of hunter-gatherer ancestry in later periods was well described before (e.g. Haak et al. 2015). Again, though, this study does add regional detail by demonstrating this resurgence to have happened later in the Baltics than Central Europe.

We argue that this is an independent finding of special interest as it explains the unique ancestry composition seen in modern Eastern Baltic populations in light of a substantial different archaeology.

In summary, while the paper presents a lot of additional regional details, I feel it lacks findings of broader interest worthy of a publication in Nature Communications.

In addition, I also have some methodological criticisms, as detailed below. However, I’m confident that the authors could address all of these in a future revision.

1) The approach to use only reads exhibiting post-mortem damage patterns to assess contamination through D-statistics is interesting. However, there are some issues with this approach. Firstly, the lack of a gene-flow signal does not necessarily imply absence of contamination. This is especially true if the test populations chosen are unlikely sources for the contamination (e.g. diverged Asian and African populations). To test for inter-sample contamination, the other samples should probably be used, or at least modern and ancient populations from the region (e.g. contemporary populations of Scandinavian, Baltic and German origin, or the samples of Skoglund et al.

2014). Secondly, PMD patterns can easily result from other sources (e.g. UV light, inter-sample contamination), hence comparing D-statistics of PMD affected reads and other reads is likely underpowered, especially if there are only few reads. Thirdly, it is unclear how many reads are actually required to detect

contamination using this approach, and I doubt that fewer reads are required than for the usually used methods. In summary, I think this approach could be an interesting methodological addition to the paper, but the authors would need to show with some artificial (bioinformatic) contamination from both other samples as well as modern Baltic sources that the amount of data they have was sufficient to detect it.

We have investigated this approach further only to find that it lacks in power, especially when the contaminating source is genetically similar to the contaminated sample. We therefore removed this analysis and now rely in our quality control on common standards: typical patterns of deamination towards the read ends, low contamination on the mtDNA, low heterozygosity on the X chromosome of males, lack of unusual ratios of reads mapping to the X and Y chromosomes, and internal consistency of genetic/archeological groups.

2) The lack of contamination estimation of the Popovo2 sample (it had too few reads for ANGSD contamination estimation, and is also missing mtDNA contamination information) makes it hard to consider this sample as uncontaminated. Given the expected abundance of mtDNA reads (e.g. off-target reads), a direct mtDNA contamination estimate should at least be attempted. As mentioned above, the D-statistics method is not sufficient unless its power is well demonstrated. Without demonstrating that this sample is uncontaminated, it must be removed from the analysis.

We have added the mtDNA contamination estimate (0–2%) for this sample.

3) There is also a major issue with the highly contaminated sample Saxtorp5158T (50-52% mtDNA contamination according to Extended Data 1). This sample was found to be related to another sample from the dataset and excluded from any further analysis because of their first-degree familial relationship. I emphasize that this sample should have been excluded on the basis of the high contamination estimate alone. Unless further information about the nature of the contamination is provided (e.g. what and how many different contaminating haplogroups; software schmutzi by Renaud et al. 2015 could be of use, maybe it was even used for the original mtDNA contamination estimates), it cannot be excluded that even the first-degree relationship between the samples could have resulted from an inter-sample contamination (undetectable via PMD filtering applied here to correct for the contamination). I therefore do not find the inference of the familial relationships of this sample valid and I suggest it is excluded from the paper. The correct reason for the exclusion of this sample is important because familial relationships could be used in interpretations of archaeological contexts.

We agree with the reviewer and have excluded the sample from the analyses

based on the high mtDNA contamination rate.

4) The method used to “call” SNPs was to randomly select a single read at each position. This can hardly be termed “calling genotypes”, and the authors should rephrase that statement. Given the often relatively high coverage, especially on targeted SNPs (up to 8.8x), it is disappointing that only a very small portion of the original information was actually used. In recent years, it become more common practice to at least select the most frequent allele to represent the position, which decreases the error rate as a function of coverage. More recently, tools have also been published that use the information from all the reads probabilistically and take into account both sequencing error rate and PMD, to call both genotypes as well as pseudo-haploids (e.g., Hofmanova et al. 2016, Link et al. 2017), and I suggest to the authors to consider using such methods in future. This is particularly relevant for studies like this one in which very different sequencing platforms and centers were used. Samples thus likely differ in their sequencing errors (both in amounts as well as type) and post-mortem damage profiles, and by sampling an allele randomly, this results in genetic distances hard to compare. While I do not request that the authors redo the complete analysis using a more appropriate calling method (especially since their results are consistent with previously published results), they should at least add to the Extended Data 1 all information regarding where and on which platform each sample was sequenced, so readers concerned with this issue might judge the consistency of the inferences and potential bias from created by the different sequencing strategies.

We have rephrased the statement in the methods section to “For each of the targeted 1240k SNP positions, a read was chosen at random to represent this position using the genotype caller pileupcaller (<https://github.com/stschiff/sequenceTools>). “. We have also added a column for the sequencing facility and platform to Supplementary Data 1.

5) The authors mention from the line 263 onwards that “We demonstrate here that early Swedish farmers, similarly, derive from a mix between Early Farmers and WHG, without additional gene-flow from foraging groups represented by SHG from Central Sweden. However, as no forager samples from South Sweden, Denmark or Northern Germany have been studied as of yet we cannot say if the genetic substrate was more WHG-like in these regions and if local admixture might have played a greater role in the TRB culture.” They also mention that they have incorporated in their analysis samples from Fu et al. 2016 (line 522) where there are among others Mesolithic WHG samples from Berry au Bac and Chaudardes (Northeast France) and Bockstein and Ofnet (from Germany). Also a sample from Loschbour, Luxembourg is available (Lazaridis et al. 2014). It is hard to assume that these less than 10,000 years old hunter-gatherers could not be very advantageously used to represent the genetic variability of WHG in these regions. It would therefore significantly add to the paper if such samples were utilised to investigate if the observed hunter-gatherer admixture

in TRB samples was of a local Baltic origin or not.

We have now included analyses using different Western, Baltic and Scandinavian HGs as source in addition to LBK and can only exclude SHG. We have therefore modified our statement to give equal weight to the possibilities of the HG component being of Southern Scandinavian origin or Central European origin, as we have no knowledge of the genetic substrate in the region before the arrival of farming.

6) The observed relationships of Mesolithic hunter-gatherers (EHG-SHG-Kunda-WHG) appear consistent with an East-West and North-South pattern of isolation by distance. Such a scenario is further supported by relatively small and step-wise differences between the samples in the ADMIXTURE analysis. The main support for these hunter-gatherer populations being distinct seems to come from the PCA. However, since all ancient DNA samples were projected on genetic variation of modern West Eurasians, the position is not very accurate and small differences between ancient samples should not be overinterpreted, especially since the samples locations are rather far apart geographically. In addition, the later Narva samples appear intermediate between the SHG and WHG groups, further questions this clear distinction. It might thus be better if samples were analysed on an individual basis where it is possible (D-statistics, qpWave), rather than pooling based on PCA coordinates.

Pooling of samples was based on PCA, admixture and individual outgroup f3-statistics. We agree that the position on the PCA should not be overinterpreted. With the addition of more Baltic Mesolithic and Early Neolithic hunter-gatherers we have analysed them individually to reveal that they indeed carry varying amounts of WHG and EHG ancestry, which explains their intermediate position between WHG and EHG, and cannot be simply due to isolation by distance when taking geographical and temporal patterns into account.

7) On the lines 237-242, the authors write “Notably, in addition to haplogroup H, the maternal lineages seen in eastern Baltic samples (n=31; Supplementary Information Figure S5) encompass all of the major haplogroups identified in complete mtDNA genomes from Holocene Scandinavian and western European hunter-gatherers (n=21:U2, U5a, U5b) 12 , as well as haplogroup U4 which has been found in high frequency in Mesolithic foragers from Russia 24 and K1, a derivative of the U8 branch found in Scandinavian foragers 20 , suggesting a large census size of Baltic hunter-gatherers to maintain such genetic diversity”. The statement on the increased population size is also present in the Conclusion (the line 418). Comparing genetic diversity based on haplogroup distributions is tricky, since haplogroups are not genetically equally distant from each other. In addition, the arbitrary pooling of haplogroups to “major haplogroups” has a huge impact on such a comparison. However, genetic diversity can be easily calculated directly from mitochondrial sequences without using haplogroups assignments. To make a statement regarding increased diversity then requires a proper comparison to a hunter-gatherer

sample encompassing a geographical region of similar size and a similar chronological transect (e.g. selection of data from Brandt et al. 2013 and Post et al. 2016).

We have removed speculations about the population size of Baltic hunter-gatherers.

8) Starting from line 242, the authors write “We see in Baltic foragers no genomic evidence of gene-flow from Central European farmers or any Y-chromosomal or mitochondrial haplogroups that are typical for them, suggesting that any traces of agriculture and animal husbandry in the Baltic Early and Middle Neolithic were due to local development or cultural diffusion.” Yet starting from the line 232: “The Narva individual Spiginas1 (dated to ca. 4440–4240 cal BCE) belongs to a mitochondrial haplogroup of the H branch providing the first direct evidence that this branch was present among European foragers without gene-flow from farmers (Table 1, Figure 2b).” and “... and K1, a derivative of the U8 branch found in Scandinavian foragers” on line 240. Using the argument of the mitochondrial haplogroup in the first statement is difficult since, as the authors mention in the second statement, some of the found haplogroups are actually not typical for hunter-gatherers (K1 and H). Indeed, in the reference (20) given, which is (Malmström et al. 2015), the haplogroup was used as an argument in favor of admixture with farmers. Interestingly, the only argument for K1 being found in hunter-gatherers of the region is found their own manuscript, namely in sample Donkalis5, which is dated very early (6075–5920 calBCE). Since this is an interesting result, also to support some of the other claims, it would be interesting to discuss the archeological interpretation for this sample. In doing so, please note that K1 has so far only been reported in pre-neolithic samples from Greece.

We have removed the first quoted sentence. Instead we discuss the presence of H and K1 as possible previously unidentified or misinterpreted branches present before the arrival of farmers, citing relevant finds from Greece (Hofmanova et al. 2016) and Romania (Gonzalez-Fortes et al. 2017).

9) Admixture was performed (as sometimes seen) at the same run with both modern and ancient populations. The definition of ancestral clusters in the analysis is therefore hindered by the heterochronous character of the data. Since the authors interpret also only very small ancestry components and differences in these between samples, it would be important 1) to check that these shared components are not artifacts of running the analysis with a huge panel of distantly related modern samples 2) to add existing ancient samples that would help in the made interpretation. For instance, to interpret the component maximised in Native Americans as a evidence for ancestry related to the MA1 and AG3 samples, these two ancient samples must be included in the analysis (I note that I did not have Extended Data Figure 4, maybe the samples were included there?). Also, bootstrapping should be used to add confidence intervals and to test whether these small components are actually meaningful.

We have performed formal tests (D-statistics) and report the results (previously in Supplementary Table S12, now found in Supplementary Tables 2 and 4) to show that EHG and SHG share an excess amount of alleles with MA1 and AG3 and do not rely simply on ADMIXTURE results. Supplementary Figure 4 now includes results from select modern individuals from the same ADMIXTURE run. Also, we have now run ADMIXTURE with 100 bootstrap replicates on 378 ancient and 3784 modern individuals from over 70 worldwide populations. For ease of illustration we chose to present only individuals relevant for our study.

Minor points:

- Extended Data Figures 1-4 were missing from the information provided. Also, Extended Data Figures 1-3 are never cited in the main text.

We have fixed the missing Figures and citations.

- Mitochondrial contamination is mentioned in the Quality control section of the methods for sample Saxtorp5158, and the mtDNA contamination values are presented in the Extended Data 1 for other samples. However, the method used to obtain these estimates was nowhere described.

The method (schmutzi, Renaud et al. 2015) was described in the Supplementary Method section and we have added it now to the main Methods.

- The authors should avoid calling the nuclear capture samples “genomes” to prevent confusion (e.g. line 135). Since these are capture data, it is only possible to refer to those as having “genome-wide data”, as is commonly done for modern captures.

We agree and refer to our data as ‘genome-wide’.

- The title “The genetic history of Northern Europe” is not very descriptive and informative about the content of the article. While it may be difficult to pinpoint an obvious major finding for a title, it could be bound at least chronologically.

We propose the alternative title: The Genetic Prehistory of the Baltic Sea Region

- On the line 165, Fig. 2a not 2A should be used. Also, for consistency only Fig. 2a (abbreviated, e.g. line 165) or Figure 2a (not abbreviated, e.g. line 532) should be used.

We fixed these errors.

- On the line 174, “its geographical vicinity” should be “its geographically vicinity”.

We fixed this.

- The coverage for the two whole-genome shotgun samples (Kunila2 and Gyvakarai1) was only reported for the targeted SNPs. It would be useful to report also genome-wide coverage of these samples, especially since a full genome with over 7x coverage is one of the highlights of the paper and valuable by itself.

We present the average coverage on a bona fide set of 1.2 million sites that is representative for the entire genome. We argue that this is a good proxy for genome-wide coverage for the shotgun-sequenced data as often different studies report this value using a variety of different masking parameters to filter out regions of the genome that are hard to map, or fail to apply these filters, thereby artificially reducing the average coverage.

Reviewer #2 (Remarks to the Author):

The paper focuses on the genetic history of Northern Europe, a key region for which there is only a limited amount of paleogenomic data. The paper reports genome-wide DNA data from 24 ancient North Europeans spanning the transition to agriculture, and the Bronze Age.

The paper's main claims are that (1) Scandinavia was settled after the retreat of the glacial ice sheets from a southern and a northern route, (2) the first Scandinavian Neolithic farmers derive their ancestry from the Anatolian farming dispersal into Europe, 1000 years earlier than has been previously reported, (3) the range of Western European Mesolithic hunter-gatherers (WHG) extended to the east of the Baltic Sea, and persisted without gene flow till around 2,900 BC, the timing of the arrival of steppe populations (Corded Ware Culture) to this region.

The manuscript is well written and easy to follow, and the figures and tables are of excellent quality.

The claims are novel and provide an important contribution to the study of the genetics of prehistoric Europeans and this paper is a valuable contribution to the field. The analytical approaches applied are excellent, and convincing (including the correct use of relevant published ancient and modern genomic data in the various analyses).

The following points need to be addressed, as I think that they will improve the manuscript and clarify a few elements.

1. Line 208: "An eastern Baltic refugium of European hunter-gatherers" . I am not convinced that the paper provides any support for a Baltic refugium and that Scandinavia was settled after the retreat of the Ice Sheets. This fact is well reported from archaeological and paleoclimatic data, but the data presented here include previously published EHG genomes (Oleni Ostrov and Popovo) as well as new Kunda Culture Mesolithic genomes from Lithuania>

the latter are dated to the 5th millennia BC, which is several millennia after the retreat of the Ice Sheets. As such, the results of the genetics of these individuals do not provide any direct support for the timing and nature of the colonization of Scandinavia. Perhaps the authors refer to genetic continuity between hunters and farmers but this is another story as the term 'refugium' is held for regions that were occupied during the LGM (and were not necessarily ice free).

We have removed our statement about the eastern Baltic as a refugium due to its specific association with the LGM. We agree that our data that postdates the retreat of the ice sheets does not provide direct evidence for the timing of Scandinavia's colonization, which we infer from the archaeological literature. We have modified our statement to remove reference of a specific timing and instead expand on the nature of colonization by two routes for which we see indirect support in our data.

2. Line 115" Here we genetically investigate.." I suggest that the authors will provide here some concrete research questions. Without these the reader is left with the feeling that the paper just explores dynamics of populations and their admixture/turnovers, but in fact I find this paper to be well structured and clear in the specific aspects which it focuses on (as is evident in the results section from the systematic tests applied). I think that this can strengthen the paper and make it easier to follow.

We have added concrete research questions to the introduction and attempt to emphasize where previous research leaves open questions. Specifically we have added following paragraph: "Here we investigate the modes of cultural and economic transitions experienced by the prehistoric populations surrounding the Baltic Sea. Were the changes seen in the Eastern Baltic Neolithic, which did not involve the introduction of agriculture, driven by contact with neighbouring groups and if so can we identify these? Was the earliest practice of farming in Southern Scandinavia a development by a local population or did it involve migration from the south? How did the unique genetic signature of modern Eastern Baltic populations come to be?"

3. Results and Discussion section, Samples and archaeological background: there are major chronological gaps between the samples. This is fine, as this is the nature of the field, but it is important to mention this as some archaeological phases are not well represented.

4. Lines 152-4: "...these samples address the open question whether the first introduction of farming around 4,000 BCE was driven by newcomers or by local groups involving later gene-flow from Central European farmers". This should be discussed above, perhaps when outlining the main questions addressed.

We have moved this research question to the introduction.

5. Line 248" Early Neolithic Migration into Scandinavia". This title is confusing. If it is Early Neolithic then why does the D statistic is comparing TRB to Middle Neolithic Central Europe, X, Mbuti? Should the MN C Europe not be replaced

by an Early Neolithic either LBK or Anatolian? It is logical to show that there is no significant results and that we can assume that TRB is closely affiliated with the MN Central Europe, but then this does not imply directly that TRB (the first introduction of agriculture to South Sweden) was the outcome of the demic diffusion. If this is the case, then it is necessary to at least compare it to LBK as MN are not early farmers and already further admixed with WHG than the first farmers. The same argument applies for eth PWC, should it not be compared to early Neolithic instead of MN? In any case there is no evidence for an actual demic diffusion. It is best to discuss the appearance of the first farmers in Scandinavia but since there is a major time delay between this event and the first demic diffusion into Europe, I think that it is better not to apply this term (also since there is no evidence for an actual logistic population growth and gradual diffusion and this is not tested here).

Early Neolithic in this case referred to the Scandinavian chronology (starting only around 4,200 BCE), which we agree might lead to confusion. We now use LBK as a source population for EN TRB in our analyses, showing that EN TRB contains increased WHG admixture when compared to LBK, which makes it indistinguishable from the European MNChL and shows no evidence of SHG admixture. We discuss the different possibilities of how this admixture pattern might have appeared in Southern Scandinavia, and refrain from using the term 'demic diffusion'.

6. Line 518: Population genetics analysis. I think that there is a sentence missing about the D statistics.

We erroneously wrote f4-statistics instead of D-statistics and have fixed the mistake.

7. Line 542: should be "...population X and Y after..."

We fixed this typo.

8. Supplementary Information Table S4. Significantly positive results for D(Ancient Eastern Baltic population, A; X, Mbuti). I think that the authors need to revise the description below it as it discusses 'modern' rather than 'ancient'.

The Table legend has been corrected accordingly.

Reviewer #3 (Remarks to the Author):

- Mittnik et al. analyse key samples from the Baltic region spanning the Mesolithic, Neolithic and Bronze age. This is a nice addition to the growing literature on ancient European genomics. Although perhaps a bit lengthy, the manuscript is well written and provides genetic support for two independent movements into Scandinavia. There are specific concerns regarding the heavy reliance on qpWave and qpAdm, and the exact interpretations of the results of these analyses. However, most conclusions can be reached without the need of these analyses. Publication is supported, perhaps with some

shortening, provided the qpWave/qpAdm issues are addressed. More specific comments follow.

- There appears to be some confusion regarding what qpWave and qpAdm are actually able to test. The link between D-statistics and qpWave is explicit in that qpWave uses a matrix of F4s, thus I believe many of the qpWave/qpAdm conclusions could be reached from D-stats alone (additional tests not shown in the manuscript will be required in some cases). D-stats are now widely understood, while qpWave and qpAdm are not. The interpretability of D-stats is also simpler, as it does not require an understanding of linear algebra, potentially broadening the readership. I strongly recommend (1) reframing the discussions by focussing on results in the D-stats, and (2) any remaining qpAdm tests (to show mixing proportions) must be justified (see below).

The reviewer makes a good point, and we have now greatly reduced the amount of qpWave/qpAdm analyses and rely more on the results from D-statistics. We agree that this likely makes the manuscript more accessible to a general readership.

- Define calBCE in Abstract, or avoid. Describe EDAR allele function.

We have changed the dates in the abstract to “before present” and added a short description of the EDAR function.

-line 139: 7,500 to 5,500 calBCE for the two Mesolithic Russians is inconsistent with Tables 1 and S1. Tables 1 and S1 suggest 5500-5000 BCE for Uz0077 and Popovo2. S11 has Uz0077 dated indirectly from associated individuals to 7450-6950 calBP and Popovo2 indirectly as 9500-7500 BP or maybe younger.

We have changed the dating to 7,500–5,000 calBCE in line with the range of dates from previous literature.

- line 174 grammar: 'its geographically vicinity'.

We have fixed this typo.

-line 185-187: My understanding is that qpWave gives a lower bound on the number of distinct ancestry components in one group (left populations) that are derived from another group (right populations). It is not clear why qpWave would be able to reject the hypothesis that SHG is formed by admixture of WHG and EHG. The qpAdm test for SHG as a linear combination of WHG and EHG gives admixture coefficients as quoted from Table S9, and this cannot be rejected. I guess the text of line 185 should say qpAdm, not qpWave? However, the validity of performing this qpAdm test has not been established by first showing that qpWave with left populations={WHG, EHG} has matrix with full rank (rank=1, i.e. two source populations). I did not see such an entry in Tables S6/S7. I note that the other criterion for this qpAdm test is that qpWave with left populations={SHG,WHG,EHG} has the same

rank (i.e. rank=1), and this is confirmed in Table S7.

We have added the necessary qpWave test suggested here to show that WHG and EHG can be distinguished as two source populations. We have also rephrased the paragraph on SHG, which we agree was confusing.

-lines 187-191: The conclusion that the crimson component in Fig 2a is maximized in modern Native American populations is not clear from Fig 2a, as no Native American nor MA1/AG3 samples are shown. Could we see the ADMIXTURE output that is the source?

ADMIXTURE results including ancient Americans and modern Native Americans are included now as Supplementary Figure 4.

- line 200: Some care needs to be taken with regard to wording and the results being referred to. Table S6 contains qpWave results, which does not do ancestry modelling. This point aside, one might expect that there should be an entry in Tables S6/S7 for qpWave with left populations={EHG,WHG} and also left populations={SHG,WHG} (in 'contrast' to the left population={Kunda,WHG}). As I noted earlier, the former entry is absent.

We have added the suggested entries to the new Supplementary Table 3.

- line 217: '[Narva] can also be accounted for by admixture of Kunda with either EHG, SHG or WHG (SI Table S6)'. I believe this is an over interpretation of the qpWave results. You have left populations={Narva, Kunda}, and qpWave cannot reject rank 0. Then you add another group to the left populations (e.g. EHG), and find qpWave is unable to reject rank 1. But this does not mean that Narva has any ancestry from the added group. There is now an additional distinct ancestry component in the left populations (derived from the outgroup populations), regardless of whether Narva has any of this newly accounted for ancestry. Looking at the separate qpWave results for NarvaEast and NarvaWest, it is clear that the most parsimonious interpretation is that NarvaWest is directly descended from Kunda, while NarvaEast has additional ancestry related to one of EHG,WHG,SHG - further, the qpAdm results for NarvaEast indicate it must be EHG or SHG and not WHG.

We agree that this analysis did not serve the intended purpose and that the paragraph was rather confusing. With the addition of new Baltic hunter-gatherer samples we opted instead to estimate EHG/WHG ancestry proportions for each individual using the qpWave/qpAdm framework, after confirming the validity of using the left populations.

line 221: Unlike for NarvaEast, the qpAdm tests for left populations = {NarvaWest, Kunda, EHG or SHG} are not valid, as qpWave could not reject rank 0 for left populations = {NarvaWest, Kunda}. I suggest that a similar interpretation may be made by contrasting D-stats of the form $D(\text{NarvaEast}, \text{Kunda}, X, \text{Mbuti})$ with $D(\text{NarvaWest}, \text{Kunda}, X, \text{Mbuti})$, where $X=\{\text{EHG,SHG}, \text{etc}\}$.

We have added D-stats of the form $D(\text{Narva, WHG}; X, \text{Mbuti})$ and $D(\text{CCC, EHG}; X, \text{Mbuti})$, testing each Narva and CCC individual separately to show that most of the individuals are not simply cladal with WHG or EHG (Supplementary Table 5). We expand on this finding by assessing the admixture coefficients of each individual using left={Narva or CCC individual, EHG, WHG} to show the varying proportions of EHG and WHG ancestry in each individual (Figure 3 and Supplementary Table 3).

-lines 254-261: D-statistics are sufficient to make the point. Having the same result from qpWave is tautological as qpWave uses an F4, which is an unnormalised D-statistic. Again, the qpAdm results are not valid as you cannot reject the more parsimonious scenario of no admixture.

We have reframed and rephrased this section according to point 5) by reviewer 1 (see above).

- line 274: qpWave analysis is redundant - you already have the D-stats.

We have removed the qpAdm analysis here and only refer to D-stats.

-lines 277-281: qpAdm tests are not justified. You should show D-stats of the form $D(\text{PWC, SHG, X, outgroup})$, where X includes more populations from your qpWave right populations.

We reran the analyses and have modified this section as follows: "Indeed, the statistic $D(\text{PWC, SHG}; X, \text{Mbuti})$ reaches weak significance when X is MN TRB ($Z=2.94$) and a two-way admixture model for PWC involving SHG and TRB farmers is not rejected by qpWave/qpAdm ($74\pm 6\%$ SHG and $26\pm 6\%$ EN TRB; Supplementary Table 3). By this model, PWC is largely genetically continuous to SHG, which is congruent with their similarities in subsistence strategies, while continuity between EN TRB and the later PWC can also be seen in archaeological assemblages³⁰ and can be attributed to contact between foraging and farming groups with gene-flow from the latter into the former."

qpWave works in a pairwise way by calculating $f_4(\text{PWC, SHG, outgroup1, outgroup2})$. We specify only Mbuti as outgroup in the D-stats but do not test every possible pair from the right pops of our qpWave/qpAdm analyses, and therefore probably miss some significant D-stats that cause the strong rejection of left=(PWC, SHG) (Supplementary Table 3).

-line 287: it is not possible to test migration directionality using any of the methods employed here. Specifically, qpAdm finds the best linear combination of source populations which accounts for the target population. It is not unthinkable that swapping the target population with one of the source populations will also give a result where the admixture coefficients are all positive.

We have removed this sentence and analysis, as D-stats have not shown this qpAdm test to be justified. We see that this paragraph was confusing, as we

were not attempting to show migration directionality but simply trying to quantify possible admixture from a source into our target population. In the case we presented swapping target and source population would not produce positive admixture coefficients. We recreated the test here to show:
Left={MN TRB, EN TRB, PWC}, rank 1 p-value: 0.787, admixture coefficients EN TRB: 0.93, PWC: 0.07; SE: 0.11
Left={PWC, EN TRB, MN TRB}, rank 1 p-value: 0.815, admixture coefficients EN TRB: 50.92 MN TRB: -49.92; SE: 29.57

-lines 307-314: I certainly prefer the language 'are consistent with' here, compared with the language of earlier paragraphs. The same conclusions ought to be reached from looking at individual D-stats of the form $D(\text{Baltic_LN_}, \text{CordedWare_Central}; X, \text{Mbuti})$, with each of the Baltic_LN_* subpopulations.

We followed the reviewers suggestion to replace qpAdm test with D-statistics which we have employed separately for each Baltic LN individual to show if they are cladal with Yamnaya or share an excess of alleles with another population $D(\text{Baltic_LN_}, \text{Yamnaya}; X, \text{Mbuti})$ (Supplementary Table 7).

-line 324: please justify this qpAdm analysis with the relevant qpWave prerequisites.

We have removed the qpAdm analysis here as the D-stats are sufficient to show excess allele sharing with Baltic EMN Narva in the youngest Baltic LN individual.

- lines 365-375: More qpWave interpretations... The cause of Baltic LN giving higher p-values over Scandinavia LNBA is apparent from the D-stats in table S5, which suggests a Levantine influence. This does not support the conclusion that Oslund is more closely related to Baltic LN, in fact the opposite appears to be true (D-stats and PCA both indicate this). Please consider carefully the relationship between qpWave results and the relevant D-stats.

We agree and have removed the qpAdm analyses and their interpretation.

- line 516: should be figure S6. It is not clear from the methods if only the shotgun data was used for sex determination, or if capture data were used also. I suspect this method would work poorly (if at all) for capture data.

We used both capture data and shotgun data for sex assignment and have added this information to the Methods section. We follow the approach presented by Fu et al. (2016), which has been shown to be very informative for capture data.

-line 542: should 'X and X' be X and Y?
We fixed this.

-line 544: should 'A and B' be X and Y?

We fixed this.

-Supplementary Tables S4,S5,S10,S11,S12,S13,S14 (D-stats tables): Please make sure the wording in the figure legends matches the column headers. I found this wording very confusing in determining if using the BABA-ABBA convention was used, or the ABBA-BABA convention.

We have reworded these Table legends to read more clearly.

-Supplementary Tables S6, S7 (qpWave tables): The column headers 'Test', 'Reference A', and 'Reference B' don't make sense in this context. Any of the left populations are mathematically interchangeable and will give the same result (likewise for the right populations).

We have renamed the Table headers. However we would like to point out that neither left nor right populations are completely interchangeable, as we have shown above for the left population list. The first population listed on either side serves as the "base" population (Supplement for Haak et al. 2015, SI10, pg 129: "We now will take T as the base population of $L = \{T, S\}$, which simplifies the algebra."). This will lead to different results when using a low coverage genome as the base population compared to a high coverage one, as missing data in the base population kills the cyclic symmetry. Using Mbuti or Mota, which are high coverage and symmetrically related to all non-Africans, as the first outgroup is generally good practice.

- Supplementary Tables S8, S9 (qpAdm tables): Many of the rows in these tables correspond to tests which are either not justified by relevant qpWave results, or are shown to be unjustified by relevant qpWave results. There are two conditions that should be satisfied to justify using qpAdm, quoting from Haak et al. SI10, pg 129:

``Thus, if T is admixed, as above, pick a set of outgroup populations O, and

1. Check, setting left populations $L = S$, and right populations O that the matrix X has full rank $n-1$.
2. Check, again setting $L = \{T, S\}$ that there is no strong evidence that the rank of X increases with the addition of T. '

We have made sure to include all relevant qpWave tests (testing rank 0 before including another populations to the left list), however in some cases rank 0 will not be rejected with a p-value close to significance level (Supplementary Table 3) while D-stats provide significant results (e.g. Supplementary Table 5). In these cases rank 1 is generally not rejected and provides a considerably higher p-value, therefore we report the result of the rank 1 test.

-Supplementary Section 3: Y haplogroup assignment has been done based largely on C-T and G-A polymorphisms, many with only 1x or 2x. These are mostly UDG half treated libraries, but as methylated cytosines deaminate to thymine it would be sensible to avoid sites in a CpG context where most

methylation will occur. I appreciate that there are multiple sites contributing to haplogroup assignment, and no conclusions are drawn regarding the more specific haplogroup subtypes.

We have added the following caveat to the section on Y haplogroup assignment (now Supplementary Note 2): "We caution that these haplogroup assignments are based partially on C-to-T and G-to-A polymorphisms at sites that may be affected by deamination."

Reviewers' comments:

Reviewer #1 (Remarks to the Author):

I'm happy to see that the authors have addressed all comments raised in my previous review adequately. I must say that I do find it a little disturbing that so many of the original claims had to be retracted based on concerns raised by either myself or my fellow reviewers. The race for impact should not compromise how careful and rigorous we conduct our science.

I also maintain that some of the apparently distinct groupings used in this manuscript, particularly of hunter-gatherers, is a bit arbitrary and maybe more the result of the geographic distribution of samples (and some wishful thinking), rather than the true distribution of genetic diversity in time and space. In my view, alternatives to explaining everything in terms of admixture between distinct groups are dismissed too easily. I accept that this is the current standard in the field, but the smaller the geographic range studied, the more its limits become apparent.

Finally, there is an obvious spelling mistake in the abstract where it reads "... settled Scandinavia via two routes Scandinavia."

Reviewer #2 (Remarks to the Author):

I went over the manuscript and the letter to the editor with all the specified replies and the corresponding changes that were made. I think that the authors did an excellent job in terms of the revision and the resubmitted manuscript should be published.

However, I provide below a list of some minor comments and suggestions:

Line 36, remove the second word ;Scandinavia'

Line 38, it will be useful to specify the absolute date after the "one thousand years earlier"

Line 38-39: "Western European hunter gatherers" refer to "WHG" or just to various European HG? In case of the latter I recommend calling it just "European hunter gatherers" as in fact WHG is present also in the east and in the south of Europe as south as Sicily (see results in Mathieson et al. 2017 bioRxiv).

Line 49: it is not so clear what "they" refer to as you are not testing all these elements (climate, innovations, resources, disease load,) it will be best to specify perhaps just cultural (as you refer to cultural units such as 'Narva' etc.) and subsistence shifts (farmers, hunter-gatherers) as the main elements examined.

I suggest to change the word 'ceramist' to "pottery" throughout

Line 101 you define a new term "LNBA". I have seen this term in the Goldberg et al. PNAS paper and it is not a term which is accepted by archaeologists even if it seems to make sense. One reason is that it skips the Copper Age which is the case for northern Europe but not for most parts of Europe. I admit that the archaeological periods and terms are confusing, but creating new ones is not the best way forward especially since your paper provides excellent archaeological context. So perhaps it is best not to use LNBA but simply to use the full "Late Neolithic/Bronze Age" throughout.

Similarly in line 332 you use the abbreviation "Baltic BA". You also use terms such as "EMN Narva" and "Baltic MN CCC", "LNBA expansions" etc. While I understand that this is related to the abbreviations

used in the statistical tests, it is still best to use full terms as much as possible when referring to specific archaeological periods so just separate between the discussion of analytical results for which you need to refer to the abbreviated parameters, and the actual periods as this will also make the paper more accessible to broader audiences.

Line 401: remove "in"

Line 402" Conditions in Scandinavia also supported the prolonged parallel existence of hunter-gatherers with farmers.." I find this sentence to be rather vague, which "conditions" and are you certain that these (environmental? Economic?) conditions supported prolonged parallel existence and that groups were not isolated from each other, perhaps with a few cases of admixture events? Total isolation will result in no admixture, while continuous or frequent contact will result in gene flow and total mixture of the source populations.

Line 412. Perhaps change " Networks of contacts" to "social networks"?

424-24" The genetic component from Levantine farmers" is actually from "NW Anatolian Neolithic farmers"

lines 426/7" I am not sure that it suggests import of a new economy but perhaps acculturation of steppe migrants into the local agricultural mode of life?

Reviewer #3 (Remarks to the Author):

The authors have addressed all of my original points. Some additional (minor) things are mentioned below.

line 36:

'Scandinavia' at end the sentence is likely a copy/paste error.

lines 200/201: 'EHG appears most significant',

line 328: 'the top hits being ...',

and line 371, 'the most significantly positive results ...'.

Careful here; the Z scores are a reflection of statistical power,

not (only) the quantity of gene flow. E.g. differences in Z scores may also

result from differences in the number of loci used for distinct tests.

line 493: Is there something missing at the start of this sentence?

Supp Table 4: the caption mentions group 'B', which should probably be 'X'.

There are a large number of acronyms and other abbreviations that may make this manuscript difficult to read for non specialists. While these are defined in the text, I needed to write out a cheat-sheet for myself while reviewing the manuscript. This is reproduced below to illustrate the problem. Please consider adding a glossary of terms.

WHG - Western Hunter-Gatherers (Iberia to Hungary, old-mesolithic)

EHG - Eastern Hunter-Gatherers (northwest Russia, mesolithic)

SHG - Scandinavian Hunter-Gatherers (Sweden, late mesolithic)

CHG - Caucasus Hunter-Gatherers

ANE - Ancient North Eurasian (e.g. MA1)

MA1 - Mal'ta boy (Siberia, upper paleolithic)
AG3 - AfontovaGora3 (Siberia, upper paleolithic)
EN TRB - Early Neolithic Funnel Beaker Culture (South Scandinavia, 4000 BCE)
MN TRB - Middle Neolithic Funnel Beaker Culture (3300 BCE)
PWC - Pitted Ware Culture (coastal hunter-gatherers, 3300 BCE?)
Narva - Eastern Baltic foragers, 4000 BCE
CCC - Combed Ceramic Culture, Eastern baltic foragers, 4000 BCE
Kunda - Eastern Baltic Mesolithic
Baltic EMN Narva - Eastern, Late Mesolithic - Middle Neolithic
Baltic MN CCC - Middle Neolithic CCC
Baltic BA - Bronze Age
Baltic LN - Late Neolithic
LNBA - Late Neolithic and Bronze Age (2570-2140 BCE)
CWC - Corded Ware Complex, Central and North-eastern Europe, 2900-2300 BCE
LBK - Linearbandkeramik culture, farmers, pre EN TRB
MNChL - Middle Neolithic and Chalcolithic, farmers, Central Europe
EMBA - Early and Middle Bronze Age

Response to Referees

Responses in cursive and marked yellow

Reviewers' comments:

Reviewer #1 (Remarks to the Author):

I'm happy to see that the authors have addressed all comments raised in my previous review adequately. I must say that I do find it a little disturbing that so many of the original claims had to be retracted based on concerns raised by either myself or my fellow reviewers. The race for impact should not compromise how careful and rigorous we conduct our science.

I also maintain that some of the apparently distinct groupings used in this manuscript, particularly of hunter-gatherers, is a bit arbitrary and maybe more the result of the geographic distribution of samples (and some wishful thinking), rather than the true distribution of genetic diversity in time and space. In my view, alternatives to explaining everything in terms of admixture between distinct groups are dismissed too easily. I accept that this is the current standard in the field, but the smaller the geographic range studied, the more its limits become apparent.

Our population labels in all cases but EHG were assigned a priori and reflect cultural affiliation and/or geographical and temporal distribution with the intention to address questions about the ancestry and diversity of the individuals within these groups. These labels were not meant to describe genetic entities, although the groupings might emerge as such in the results of our analyses.

We confident that our wording reflects this line of argument, and have added a sentence (187-189) to illustrate why the Mesolithic samples from Russia were grouped as EHG.

In lines 241-246 we address the competing hypotheses of a long-standing geographic cline vs. recent admixture between distinct groups and why the latter seems more plausible in this case. We have changed the wording to make this clearer.

Finally, there is an obvious spelling mistake in the abstract where it reads "...settled Scandinavia via two routes Scandinavia."

We have fixed this.

Reviewer #2 (Remarks to the Author):

I went over the manuscript and the letter to the editor with all the specified replies and the corresponding changes that were made. I think that the authors did an excellent job in terms of the revision and the resubmitted manuscript should be published.

However, I provide below a list of some minor comments and suggestions:

Line 36, remove the second word ;Scandinavia'

We have fixed this.

Line 38, it will be useful to specify the absolute date after the "one thousand years earlier"

We are constrained by the abstract word limit, but the absolute dates are mentioned later in the main text.

Line 38-39: Western European hunter gatherers" refer to "WHG" or just to various European HG? In case of the latter I recommend calling it just "European hunter gatherers" as in fact WHG is present also in the east and in the south of Europe as south as Sicily (see results in Mathieson et al. 2017 bioRxiv).

We changed the wording to read "Mesolithic Western hunter-gatherers", as we are referring to the "WHG" cluster of hunter-gatherers, as opposed to EHG.

Line 49: it is not so clear what "they" refer to as you are not testing all these elements (climate, innovations, resources, disease load,) it will be best to specify perhaps just cultural (as you refer to cultural units such as 'Narva' etc.) and subsistence shifts (farmers, hunter-gatherers) as the main elements examined.

"They" in this sentence was meant to refer to the "three major prehistoric migrations" of the previous sentence without any claim that we are testing all the different factors mentioned. We modified the paragraph to read less confusingly.

I suggest to change the word 'ceramist" to "pottery" throughout

We have changed "ceramist" to "pottery-producing" throughout.

Line 101 you define a new term "LNBA". I have seen this term in the Goldberg et al. PNAS paper and it is not a term which is accepted by archaeologists even if it seems to make sense. One reason is that it skips the Copper Age which is the case for northern Europe but not for most parts of Europe. I admit that the archaeological periods and terms are confusing, but creating new ones is not the best way forward especially since your paper provides excellent archaeological context. So perhaps it is best not to use LNBA but simply to use the full "Neolithic/Bronze Age" throughout.

Similarly in line 332 you use the abbreviation Baltic BA". You also use terms such as "EMN Narva " and "Baltic MN CCC", "LNBA expansions" etc. While I understand that this is related to the abbreviations used in the statistical tests, it is still best to use full terms as much as possible when referring to specific archaeological periods so just separate between the discussion of analytical

results for which you need to refer to the abbreviated parameters, and the actual periods as this will also make the paper more accessible to broader audiences.

As suggested by reviewer #3, we have added a glossary of abbreviated archaeological periods and cultures in Supplementary Note 1 that we hope will improve the readability and accessibility of the manuscript. Generally, we have chosen abbreviations that are commonly used and accepted in archaeogenetics papers (such as LNBA introduced in Haak et al. 2015). Furthermore, we have now revised instances in the text where we felt full terms instead of abbreviations improved readability.

Line 401: remove “in”

We have fixed this.

Line 402” Conditions in Scandinavia also supported the prolonged parallel existence of hunter-gatherers with farmers..” I find this sentence to be rather vague, which “conditions” and are you certain that these (environmental? Economic?) conditions supported prolonged parallel existence and that groups were not isolated from each other, perhaps with a few cases of admixture events? Total isolation will result in no admixture, while continuous or frequent contact will result in gene flow and total mixture of the source populations.

We have changed the paragraph to include the reviewer’s points.

Line 412. Perhaps change “ Networks of contacts” to “social networks”?

We have changed this.

424-24” The genetic component from Levantine farmers” is actually from “NW Anatolian Neolithic farmers”

We have changed this.

lines 426/7” I am not sure that it suggests import of a new economy but perhaps acculturation of steppe migrants into the local agricultural mode of life?

The import of pastoralism through a migration into the region is the most parsimonious explanation, as there is no data to support widespread local agriculture/animal husbandry in the Eastern Baltic before the appearance of the Yamnaya-like genetic component.

Reviewer #3 (Remarks to the Author):

The authors have addressed all of my original points. Some additional (minor) things are mentioned below.

line 36:

'Scandinavia' at end the sentence is likely a copy/paste error.

We have fixed this.

lines 200/201: 'EHG appears most significant',

line 328: 'the top hits being ...',

and line 371, 'the most significantly positive results ...'.

Careful here; the Z scores are a reflection of statistical power, not (only) the quantity of gene flow. E.g. differences in Z scores may also result from differences in the number of loci used for distinct tests.

We agree that the highest Z score does not necessarily reflect highest gene flow, but may be affected by coverage and sample size. We now sort Supplementary Tables 2, 4, 5 and 7 by D score, and changed the sentences in the three mentioned locations (and in line 372) to reflect this.

line 493: Is there something missing at the start of this sentence?

A paragraph was shifted into the wrong section, we fixed this.

Supp Table 4: the caption mentions group 'B', which should probably be 'X'.

We have fixed this.

There are a large number of acronyms and other abbreviations that may make this manuscript difficult to read for non specialists. While these are defined in the text, I needed to write out a cheat-sheet for myself while reviewing the manuscript. This is reproduced below to illustrate the problem. Please consider adding a glossary of terms.

WHG - Western Hunter-Gatherers (Iberia to Hungary, old-mesolithic)

EHG - Eastern Hunter-Gatherers (northwest Russia, mesolithic)

SHG - Scandinavian Hunter-Gatherers (Sweden, late mesolithic)

CHG - Caucasus Hunder-Gatherers

ANE - Ancient North Eurasian (e.g. MA1)

MA1 - Mal'ta boy (Siberia, upper paleolithic)

AG3 - AfontovaGora3 (Siberia, upper paleolithic)

EN TRB - Early Neolithic Funnel Beaker Culture (South Scandinavia, 4000 BCE)

MN TRB - Middle Neolithic Funnel Beaker Culture (3300 BCE)

PWC - Pitted Ware Culture (coastal hunter-gatherers, 3300 BCE?)

Narva - Eastern Baltic foragers, 4000 BCE

CCC - Combed Ceramic Culture, Eastern baltic foragers, 4000 BCE

Kunda - Eastern Baltic Mesolithic

Baltic EMN Narva - Eastern, Late Mesolithic - Middle Neolithic

Baltic MN CCC - Middle Neolithic CCC

Baltic BA - Bronze Age

Baltic LN - Late Neolithic

LNBA - Late Neolithic and Bronze Age (2570-2140 BCE)
CWC - Corded Ware Complex, Central and North-eastern Europe, 2900-2300
BCE
LBK - Linearbandkeramik culture, farmers, pre EN TRB
MNChL - Middle Neolithic and Chalcolithic, farmers, Central Europe
EMBA - Early and Middle Bronze Age

*We have added a glossary of terms and abbreviations as Supplementary
Note 1.*

REVIEWERS' COMMENTS:

Reviewer #1 (Remarks to the Author):

I'm happy to see that the authors now mention isolation by distance as an alternative explanation for the pattern of hunter-gatherer ancestry in Narva individuals. This is an exciting period and I'm looking forward to future studies focusing on unbiased markers that will hopefully allow us to describe the demographic processes in more detail.

I have further comments.

Reviewer #2 (Remarks to the Author):

I have read the reply to reviewers letter and the resubmission and I am fully satisfied with the changes that were made as the authors address each and every point. .

I recommend to accept this resubmitted version of the publication